# Light-dependent N-end rule-mediated disruption of protein function in *Saccharomyces cerevisiae* and *Drosophila melanogaster*

**Leslie M. Stevens**[1⊙], **Goheun Kim**[1⊙], **Theodora Koromila**[2⊙], **John W. Steele**[1], **James McGehee**[2], **Angelike Stathopoulos**[2]*, **David S. Stein**[1]*

**1** Department of Molecular Biosciences and Institute for Molecular and Cellular Biology, The University of Texas at Austin, Austin, Texas, United States of America, **2** Division of Biology and Biological Engineering, California Institute of Technology, Pasadena, California, United States of America

⊙ These authors contributed equally to this work.
* angelike@caltech.edu (AS); d.stein@mail.utexas.edu (DSS)

**Data Availability Statement:** All relevant data are within the manuscript and its Supporting Information files.

## Abstract

Here we describe the development and characterization of the photo-N-degron, a peptide tag that can be used in optogenetic studies of protein function *in vivo*. The photo-N-degron can be expressed as a genetic fusion to the amino termini of other proteins, where it undergoes a blue light-dependent conformational change that exposes a signal for the class of ubiquitin ligases, the N-recognins, which mediate the N-end rule mechanism of proteasomal degradation. We demonstrate that the photo-N-degron can be used to direct light-mediated degradation of proteins in *Saccharomyces cerevisiae* and *Drosophila melanogaster* with fine temporal control. In addition, we compare the effectiveness of the photo-N-degron with that of two other light-dependent degrons that have been developed in their abilities to mediate the loss of function of Cactus, a component of the dorsal-ventral patterning system in the *Drosophila* embryo. We find that like the photo-N-degron, the blue light-inducible degradation (B-LID) domain, a light-activated degron that must be placed at the carboxy terminus of targeted proteins, is also effective in eliciting light-dependent loss of Cactus function, as determined by embryonic dorsal-ventral patterning phenotypes. In contrast, another previously described photosensitive degron (psd), which also must be located at the carboxy terminus of associated proteins, has little effect on Cactus-dependent phenotypes in response to illumination of developing embryos. These and other observations indicate that care must be taken in the selection and application of light-dependent and other inducible degrons for use in studies of protein function *in vivo*, but importantly demonstrate that N- and C-terminal fusions to the photo-N-degron and the B-LID domain, respectively, support light-dependent degradation *in vivo*.

**Funding:** This work was funded by grants from the NIH Office of the Director (https://www.nih.gov/institutes-nih/nih-office-director) R21OD017964 (awarded to D.S.S., supporting L.M.S., G.K., and D.S.S.), the National Institute of General Medical Sciences (https://www.nigms.nih.gov) R35GM118146 (awarded to A.S., supporting T.K. and A.S.), and the Eunice Kennedy Shriver National Institute of Child Health and Human Development (https://www.nichd.nih.gov) R21HD095639 (awarded to A.S., supporting J.M. and A.S.) of the National Institutes of Health. J.W.S. was funded by the Cell and Molecular Biology Graduate Program and the Institute for Cellular and Molecular Biology at the University of Texas at Austin. The funders had no role in study design, data collection and analysis, decision to publish, or preparation of the manuscript.

**Competing interests:** The authors have declared that no competing interests exist.

## Author summary

Much of what we know about biological processes has come from the analysis of mutants whose loss-of-function phenotypes provide insight into their normal functions. However, for genes that are required for viability and which have multiple functions in the life of a cell or organism one can only observe mutant phenotypes produced up to the time of death. Normal functions performed in wild-type individuals later than the time of death of mutants cannot be observed. In one approach to overcoming this limitation, a class of peptide degradation signals (degrons) have been developed, which when fused to proteins-of-interest, can target those proteins for degradation in response to various stimuli (temperature, chemical agents, co-expressed proteins, or light). Here we describe a new inducible degron (the photo-N-degron or PND), which when fused to the N-terminus of a protein, can induce N-end rule-mediated degradation in response to blue-light illumination and have validated its use in both yeast and *Drosophila* embryos. Moreover, using the *Drosophila* embryonic patterning protein Cactus, we show that like the PND, the previously-described B-LID domain, but not the previously-described photosensitive degron (psd), can produce detectable light-inducible phenotypes in *Drosophila* embryos that are consistent with the role of Cactus in dorsal-ventral patterning.

## Introduction

More than a century of genetic analysis underlies much of our understanding of biology. Unbiased genetic screens utilizing chemical mutagens, ionizing radiation or insertional mutagenesis with transposons or retroviruses, and more recently, reverse genetic strategies capable of generating precisely targeted mutations, have been critical in uncovering the genes, proteins and mechanisms underlying normal physiology as well as the processes that go awry in various disease states. However, for genes with products that are required early in the life of an organism, it can be challenging to generate loss-of-function mutant individuals in which later phenotypes associated with protein loss can be examined. While the use of site-specific recombination systems to generate clones of cells lacking expression of a protein in the background of an otherwise viable individual [1–4] can, in some cases, overcome this barrier, proteins already present may perdure for some time and even multiple cell generations after mutant clone induction, which can complicate the analysis of the loss-of-function phenotypes. This is especially problematic in situations in which it is desirable to achieve rapid protein inactivation, such as investigations of protein function at specific stages of the cell cycle, during cell migration and morphogenesis, or during neuronal signaling. Moreover, for genes encoding proteins that are necessary for cell viability, cell death following the generation of mutant clones can obscure the detection and analysis of more subtle phenotypes resulting from protein loss-of-function. Similarly, RNA interference via the expression of dsRNA or siRNAs, which has been used to interrogate the function of vital genes in a cellular or tissue-specific manner [5,6] often achieves only partial elimination of the protein-of-interest and is also susceptible to the problem of protein perdurance noted above.

Another approach to the study of proteins with functions essential for organismal or cell viability is the use of temperature-sensitive (TS) mutations. However, although methods for the rational design of TS alleles encoding proteins-of-interest exist [7–10], these approaches are associated with drawbacks that can limit their general applicability. Dohmen et al. [11] devised a general approach for expressing TS proteins, relying upon the N-end rule pathway for ubiquitin/proteasome-mediated degradation [12–14], which degrades proteins bearing N-

terminal amino acid residues other than methionine under the control of ubiquitin ligases known as UBRs or N-recognins [15,16]. Dohmen et al. [11] showed that a peptide comprising a TS version of the mouse dihydrofolate reductase protein carrying an N-terminal arginine (the temperature dependent or "td" degron) could, when fused to the amino terminus of a several yeast proteins, render the resulting fusion proteins inactive and lead to their degradation at 37˚ C but not at 23˚ C. This TS phenotype was dependent upon UBR1, the yeast N-recognin [12,15,17]. The td degron has been used to investigate protein function in a number of systems including budding yeast, fission yeast and vertebrate tissue culture cells [18–23]. Many organisms cannot survive at 37˚ C, the temperature at which the td degron mediates protein degradation. Accordingly, the low temperature-controlled (lt) degron [24], which operates over a lower temperature range (16˚C to 29˚C), was generated by modifying the td degron, which expands the organisms in which this method can be applied. However, TS mutants, including the td and lt degrons cannot be used to study protein function in homeothermic organisms such as mammals, which maintain a constant internal temperature.

A number of other degrons have been developed, which induce the proteins to which they have been attached to undergo degradation in response to addition of small molecules such as the rapamycin analogue Shield-1 [25,26] or auxin [27–29]. These approaches are highly dependent upon the extent and rapidity with which the small molecule can be administered to or depleted from the target cell/tissue/organism. Furthermore, the auxin system as well as several other inducible degron systems require the co-expression of a heterologous specificity-conferring factor together with the degron-tagged target protein [30–33]. While these approaches can provide tissue specificity, based on the expression pattern of the specificity conferring factors, they are obviously influenced and potentially limited in utility by the time required for induction and expression or loss of these factors.

In recent years, a number of novel strategies have been developed in which light is employed to modulate protein behavior in powerful new approaches to examining biological processes. These new technologies, which comprise the rapidly expanding area of optogenetics [34,35], have revolutionized several realms of biomedical research, leading to the expression of light-sensitive membrane channels [36–38] and the generation of proteins that undergo light-dependent conformational changes that affect their activity [39–41], cellular localization [42–44], and protein-protein interactions [45,46].

We began our studies intending to combine recent advances in the understanding of light-modulated proteins and of ubiquitin/proteasome-mediated protein degradation to develop techniques that permit light-activated degradation of target proteins. In addition to overcoming the limitations associated with the conditional approaches described above, such a system could potentially enable a level of temporal and spatial precision not possible using currently available systems for perturbing gene expression. Many of the gene products required for correct embryonic development in *Drosophila* are expressed maternally and deposited as mRNAs or proteins into the developing egg during oogenesis, for later function in the embryo. For those gene products required for viability of the female fly, or during oogenesis to produce an egg, it is difficult to examine the phenotypic consequences of loss of function of proteins produced maternally that are required to support progeny embryogenesis. Thus, the ability to rapidly eliminate otherwise stable proteins with the application of light could provide a valuable tool for the examination of loss-of-function phenotypes whose visualization would be precluded by the perturbation of earlier loss-of-function phenotypes.

Proteins containing LOV domains respond to environmental stimuli such as Light, Oxygen and Voltage by undergoing conformational changes [47,48]. Light-dependent members of this class of proteins utilize flavin cofactors as chromophores to function as blue-light sensitive photoreceptors in bacteria [49], fungi [50,51] and plants [52]. Structural studies of one of the

two LOV domains present in phototropin 1 of *Avena sativa* (common oat) (phLOV2) showed that under blue light illumination, the flavin-binding region dissociates from and unwinds an adjacent alpha helical region termed Jα [53–55]. Other studies have recently established that proteins bearing modified LOV domains can be induced to degrade in response to light [56,57].

Here we report on the development of an additional light-dependent degron that makes use of the phLOV2 domain from *Avena sativa* (oat), which we term the photo-N-degron (PND). When attached to the N-terminus of several proteins which are then expressed in yeast, the PND induces light-dependent N-end rule-mediated loss of function, owing to protein degradation. We also show that in *Drosophila* embryos, the PND can induce a rapid light-dependent loss of Cactus, the cognate inhibitor of Dorsal, the fly orthologue of the mammalian transcription factor NFκB [58]. Cactus and Dorsal are components of the signal transduction pathway that defines *Drosophila* embryonic dorsal-ventral (DV) polarity [59] and light-induced degradation of PND-tagged Cactus leads to alterations in DV patterning. Finally, we compare the abilities of the psd [56], the B-LID domain [57], and the PND, finding that, like the PND, the B-LID domain directs robust elimination of Cactus function, while the psd degron leads to only a subtle phenotypic difference upon illumination. Our results demonstrate that the PND can be a powerful tool for conditional elimination of proteins-of-interest for phenotypic studies *in vivo* and stress the need for care in the selection of engineered degrons for use in studies of protein function.

## Results

### The LOV2 domain from plant phototropin 1 bearing an N-end rule targeted arginine domain directs light dependent loss-of-function phenotypes in yeast

**a. Ura3p.** As noted above, proteins carrying LOV-sensitive domains respond to various environmental stimuli by undergoing conformational changes [47,48]. We reasoned that if properly positioned at the amino terminus of a protein-of-interest, light-dependent unwinding of the Jα helix within the LOV domain could act analogously to the temperature-dependent unfolding of DHFR[ts] to facilitate degradation of the fusion protein by the N-end rule degradation pathway. Proteins with atypical N-terminal amino acid can be generated experimentally by expressing the protein-of-interest as in-frame fusions to the C-terminus of ubiquitin. Because the ubiquitin monomer is cleaved co-translationally through the action of a deubiquitinating enzyme [11,12,60], it does not mark the protein for proteasomal degradation and the amino acid immediately following the ubiquitin becomes the N-terminal residue of the fully translated protein.

Accordingly, we engineered constructs that would generate a protein bearing an N-terminal ubiquitin moiety followed by an arginine residue (R), which would correspond to the amino terminus after co-translational removal of ubiquitin. The arginine was followed by the 144 amino acid LOV2 domain of phototropin 1 from *Avena sativa* (phLOV2), a single in-frame copy of the HA epitope [61] and finally the coding sequence of the yeast orotidine-5'-phosphate decarboxylase protein, Ura3p, which is encoded by the *URA3* gene. The basic structure of this Ubi-R-phLOV2-HA-Ura3p fusion protein and how it is presumed to direct light-inducible degradation of Ura3p is depicted in Fig 1A and 1L, respectively. Because it was not known whether the addition of the R-phLOV2 element would render the fusion proteins too labile or too stable to detect phenotypic differences under dark versus illuminated conditions, we constructed additional plasmids to express versions of the protein in which putative stabilizing or destabilizing stretches of amino acids, from DHFR and DHFR[ts], respectively [11],

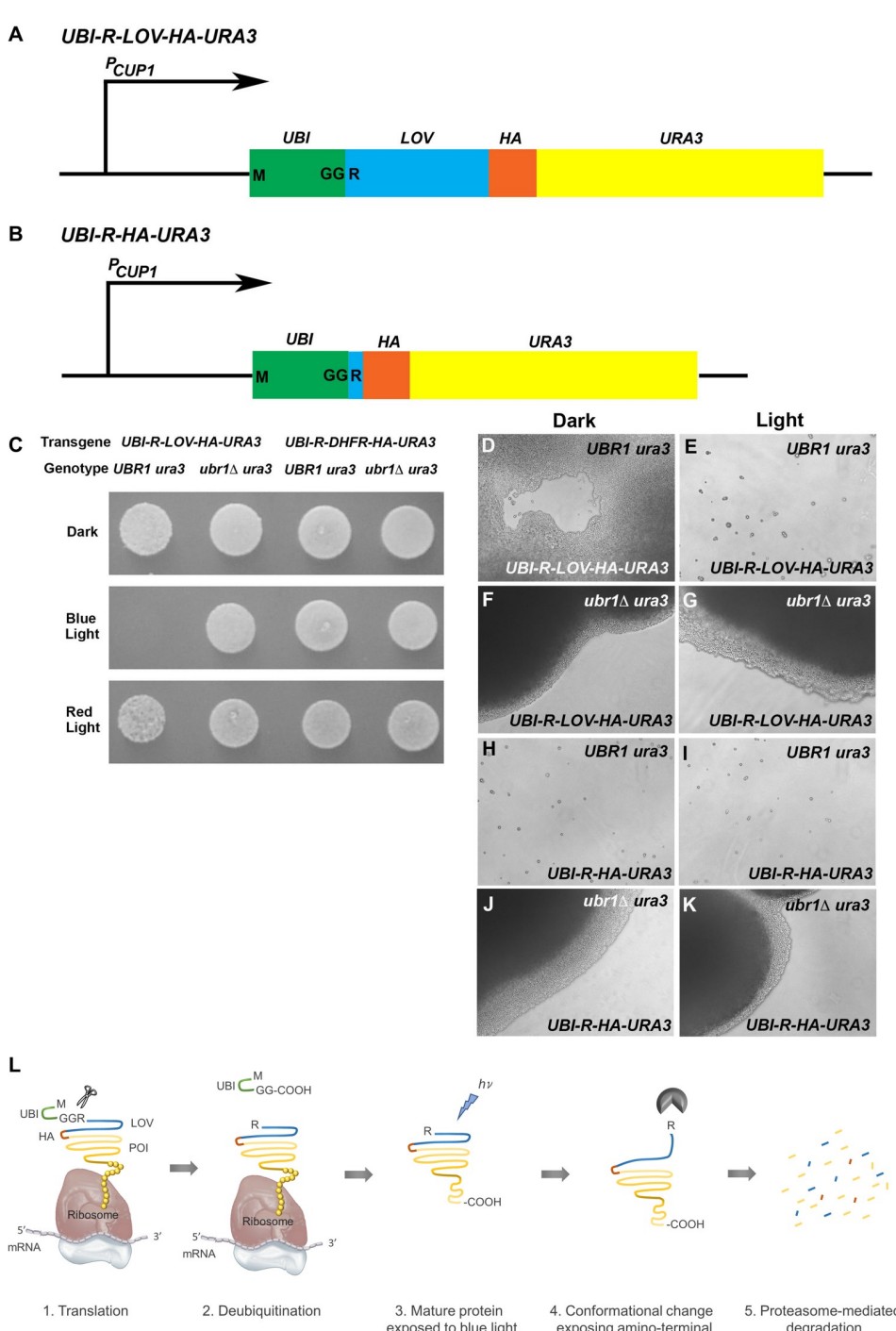

**Fig 1. An amino terminal domain encoding ubiquitin, fused to the LOV2 domain from oat phototropin I mediates blue light/N-end rule-mediated loss of Ura3p function in yeast.** (A) A schematic diagram showing the organization of the construct encoding blue/light, N-end rule targeted Ura3p, under the transcriptional control of the copper-inducible *CUP1* promoter ($P_{CUP1}$). From 5' to 3', the transgene encodes a single copy of the ubiquitin open reading frame (*UBI*), the LOV2 domain from plant phototropin I (*LOV*), a single copy of the influenza hemagglutinin epitope (*HA*), and the open reading frame encoding the yeast Ura3p protein (*URA3*). Protein synthesis initiates at the ubiquitin initiation codon (M) and the pair of glycine residues at the C-terminus of the ubiquitin open reading frame (GG) are followed immediately by an arginine codon (R). The ubiquitin domain is removed co-translationally, leaving the arginine residue immediately preceding the LOV domain as the N-terminal residue of the mature protein. In the corresponding *UBI-R-DHFR-HA-URA3* construct, the sequence encoding the LOV2 domain have been replaced by

DHFR coding sequences bearing an N-terminal arginine residue. **(B)** A schematic diagram of *UBI-R-HA-URA3*, which lacks the sequences encoding the light-sensitive LOV domain. **(C)** The *UBI-R-LOV-HA-URA3* and *UBI-R-DHFR-HA-URA3* transgenes were introduced into *UBR1 ura3* and *ubr1Δ ura3* mutant cells (introduced transgenes and yeast genotypes shown at top of panel), which were seeded onto selective plates lacking uracil and incubated in either darkness, under blue-light, or under red-light illumination. Under blue light, the *UBI-R-LOV-HA-URA3* construct failed to restore growth in the absence of uracil, indicating the sensitivity of the expressed R-phLOV2-HA-Ura3p protein to blue light. When incubated under blue light in the absence of the Ubr1p activity (*ubr1Δ ura3*), growth in the absence of uracil was restored. In contrast to R-phLOV2-HA-Ura3p, the R-DHFR-HA-Ura3p protein did not confer light sensitivity upon growth in the absence of uracil. **(D-K)** Yeast cells expressing either *UBI-R-phLOV2-HA-URA3* **(D-G)** or *UBI-R-HA-URA3* **(H-K)** (transgenes shown at bottom of panels) were expressed in either Ubr1p-expressing (*UBR1 ura3*) **(D, E, H, I)** or Ubr1p-lacking (*ubr1Δ ura3*) **(F, G, J, K)** genetic backgrounds and incubated on selective plates lacking uracil either in darkness **(D, F, H, J)** or under blue light illumination **(E, G, I, K)**. In the presence of Ubr1p and incubated under blue light illumination **(E)**, *UBI-R-phLOV2-HA-Ura3p*-expressing cells arrested mainly as single cells, arguing that the light/Ubr1p-mediated loss of Ura3p protein function was rapid. In contrast, when grown in the dark **(D)** or in a *ubr1Δ* mutant background **(G)**, these cells proliferated normally. In contrast, cells bearing a wild-type *UBR1* gene and expressing *UBI-R-HA-URA3*, arrested as single cells both in darkness and under illumination **(H, I)**, indicating that the presence of an N-end rule targeted arginine, in the absence of the phLOV2 domain, rendered the encoded protein functionally inactive regardless of light conditions **(G, H)**, while the absence of the Ubr1p ubiquitin ligase protein left the R-HA-Ura3p protein functional in darkness and under blue light illumination **(J, K)**. **(L)** shows a schematic representation of the mechanism through which R-phLOV-HA-tagged protein is presumed to be synthesized and degraded in response to blue-light illumination.

were inserted between the R-phLOV2 domain and Ura3p. All of the constructs were generated using the plasmid backbone of pPW17R [11], a yeast centromere plasmid that expresses introduced genes under the control of the *CUP1* copper-inducible promoter. However, because all three of these constructs behaved identically in the tests outlined below, only the results obtained in studies of the construct expressing U-R-phLOV2-HA-Ura3p, without additional DHFR or DHFR$^{ts}$ sequences, are described below and shown in Fig 1.

The plasmid encoding Ubi-R-phLOV2-HA-Ura3p was introduced into YPH500, a *UBR1 ura3* mutant strain [62], and into its (Ubr1/N-recognin-lacking) mutant derivative, JD15 [11]. YPH500 and JD15 are the Ubr1-expressing and Ubr1-lacking strains used in all of the yeast studies in this work. In addition to the *ura3* mutation, this strain carries additional nutritional mutations enabling selection for the presence of plasmids introduced into these two yeast strains. The full genotypes of the two yeast strains are shown in the Materials and Methods section. The abilities of the introduced Ubi-R-phLOV2-HA-Ura3p plasmid to restore a Ura+ phenotype to cells grown in the dark or under blue light illumination were examined. The plasmid conferred a Ura+ phenotype on *UBR1 ura3* cells plated in the dark (Fig 1C, top row, first yeast patch), but failed to rescue the *ura3* mutant phenotype when the cells were grown under blue light (Fig 1C, middle row, first yeast patch), provided that the yeast had initially been seeded on the plate in a single layer. When viewed microscopically, a Ura+ phenotype (growth) was observed in the dark (Fig 1D), but, in contrast, blue light illumination led to arrest of growth, in many cases as single cells (Fig 1E). When introduced into the *ubr1Δ ura3* yeast strain, the plasmid conferred a Ura+ phenotype in both dark and blue (and red) light conditions (Fig 1C, all rows, second yeast patch), indicating that the Ubr1 ubiquitin ligase protein is required for blue light-dependent loss of Ura3p activity. To show that the light-dependence was dependent on the presence of the LOV domain, we also tested a plasmid encoding Ubi-R-DHFR-HA-Ura3p [11], which conferred a Ura+ phenotype that was not light dependent, in both *UBR1 ura3* and *ubr1Δ ura3* yeast strains (Fig 1C, all rows, yeast patches 3 and 4). The flavin-containing chromophore that elicits the light-dependent conformational change in the phLOV2 domain absorbs blue light specifically. To confirm that the light-dependent Ura- phenotype was specific to blue light, we grew *UBR1 ura3* cells bearing the Ubi-R-

phLOV2-HA-Ura3p construct under red light and showed that they exhibited a Ura+ pheno-type (Fig 1C, bottom row, first yeast patch).

As an additional test of the role of the *A. sativa* LOV2 domain (i.e. phLOV2) in the light-dependence of Ura+/- phenotypes observed above, we also generated a plasmid encoding Ura3p with an N-terminal arginine residue and HA tag, but lacking the phLOV2 domain (Fig 1B). When introduced into *UBR1 ura3* host cells and placed under selection for synthesis of uracil, a Ura- phenotype (no growth) was observed in both dark conditions and under blue light illumination (Fig 1H and 1I). Conversely, when introduced into *ura3 ubr1Δ* cells, the cells grew robustly in the absence of added uracil in both the dark and under blue-light illumi-nation (Fig 1J and 1K). These observations strongly suggest that in the absence of the phLOV2 domain, the presence of a simple N-end rule-targeted amino terminus renders the Ura3p pro-tein too unstable to support the synthesis of uracil. In the dark, the presence of the phLOV2 domain immediately C-terminal to an amino terminal arginine residue stabilizes Ura3p against constitutive degradation by the N-end rule pathway. Upon blue light illumination, the phLOV2 domain presumably unfolds and this stabilizing effect is lost.

We also generated constructs analogous to the ones described above in which the LOV-domain-containing *Neurospora crassa* circadian clock regulator Vivid [50,51] or its LOV domain alone (vvdLOV), both bearing N-terminal arginine residues, substituted for the phLOV2 domain from *A. sativa* carrying an amino terminal arginine. However, as the ability of these constructs to supply Ura3p activity did not differ under blue light versus darkness, they were not pursued further in these studies.

The results outlined above, as well as data to be described below, demonstrate that the Ubi-R-phLOV2 cassette represents a transferrable element that, when attached to heterologous proteins at their N-terminus, can confer rapid, blue light- and Ubr1/N-recognin-dependent N-end rule mediated degradation. This degradation is sufficient to produce a loss-of-function phenotype. This forms the basis for naming the element the **photo-N-degron** (**PND**). Fig 1L shows a schematic representation of the envisioned process through which the PND bearing an N-terminal arginine residue is generated and leads to degradation of protein to which it has been fused.

**b. yEmRFP.** In order to explore the ability of the PND cassette to direct the loss of other proteins, we added it in-frame to the amino terminus of yEmRFP [63] (Fig 2A), a version of the mCherry mRFP variant that is optimized for yeast codons. As described above for Ura3p, this construct was carried on the plasmid backbone obtained from the pPW17R yeast centro-mere plasmid and expressed in the YPH500 (*UBR1*) and JD15 (*ubr1Δ*) strains under blue light illumination and in darkness. As a control, a construct expressing N-end rule-targeted Arg-yEmRFP lacking the phLOV2 domain was expressed under the same conditions (Fig 2B). As the yEmRFP protein confers no selective advantage or disadvantage upon yeast cells, patches of *UBR1* and *ubr1Δ* cells expressing either PND-yEmRFP or Arg-yEmRFP grew up robustly under either blue light illumination or in darkness.

*UBR1* cells expressing PND-yEmRFP exhibited easily detectable levels of red fluorescence when grown in the dark (Fig 2C). In contrast, under blue light illumination the fluorescence was almost undetectable (Fig 2D). In the *ubr1Δ* mutant strain, there was no significant differ-ence in the fluorescence levels between dark and blue light conditions (Fig 2E and 2F). Thus, PND-yEmRFP exhibited blue light and UBR1-dependent loss of fluorescence. No difference in fluorescence was observed between dark and blue light conditions in yeast expressing R-yEmRFP (Fig 2G and 2H), indicating that the presence of the PND cassette was responsible for the light-dependent effect seen in Fig 2C and 2D. Similarly, when grown in the *ubr1Δ* strain there was no light-dependent change in fluorescence associated with R-yEmRFP (Fig 2I and 2J). It is notable, however, that in the *UBR1* background, cells expressing R-yEmRFP exhibited

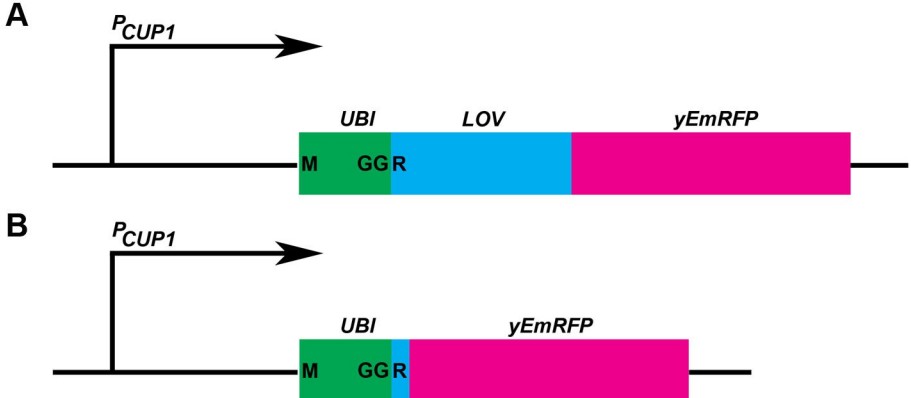

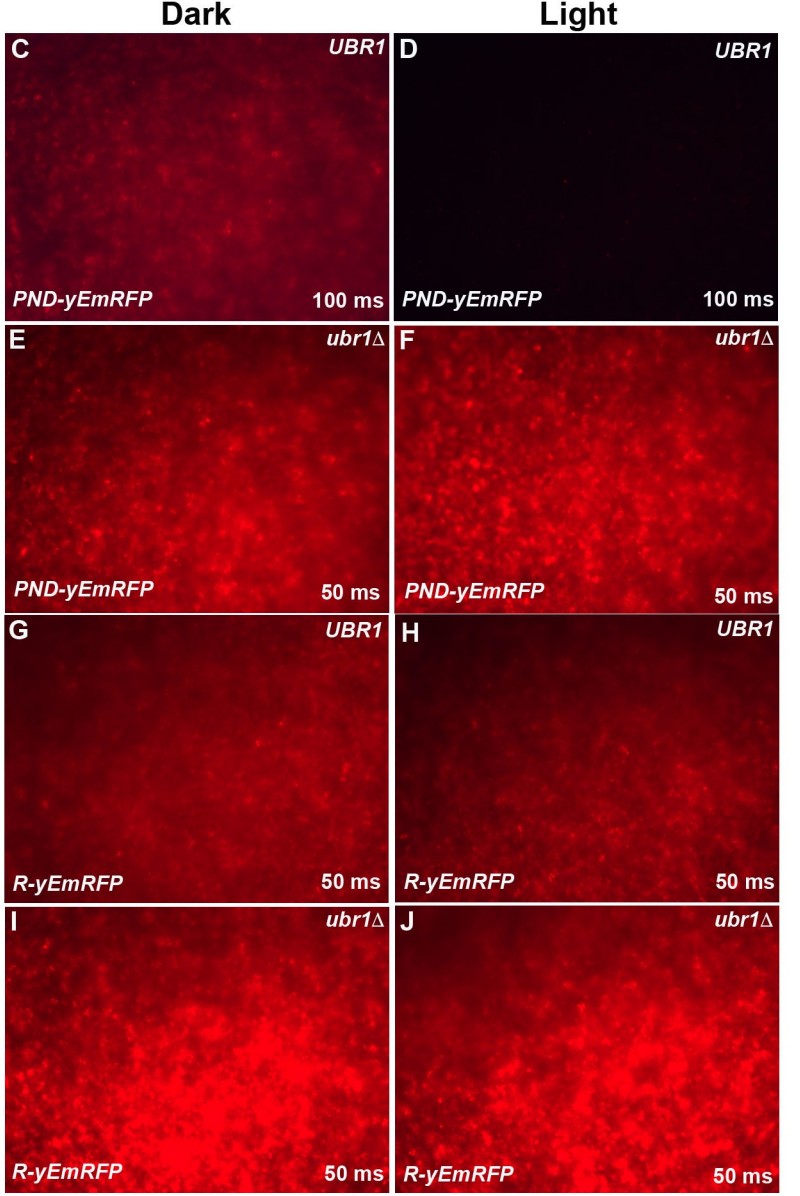

**Fig 2. The PND mediates blue light/Ubr1-dependent loss of yEmRFP-associated fluorescence in yeast.** Two constructs encoding yEmRFP bearing either the PND (*PND-yEmRFP*) (**A, C, D, E, F**) or a single arginine (R) residue (*R-yEmRFP*) (**B, G, H, I, J**) at the amino terminus were introduced into *UBR1* (**C, D, G, H**) and *ubr1Δ* (**E, F, I, J**) strains of yeast (labelled at top right of each panel) and seeded onto selective plates. Following 48 hours growth to confluence in darkness (**C, E, G, I**) or under blue light illumination (**D, F, H, J**) the surfaces of the patches were imaged for red fluorescence. Photographic imaging was carried out on the same day and under the same conditions, with the time of exposure (50 or 100 milliseconds [ms], noted at bottom right of each panel) the same for each of the light/dark pairings. This permitted a determination of relative levels of expression between light/dark pairings and between the strains and the constructs that they carried. Yeast expressing *PND-yEmRFP* in the presence of Ubr1p, exhibited a dramatic decrease in response to illumination (**C, D**). In the absence of Ubr1p, yeast bearing this construct expressed higher levels of fluorescence that were not affected by illumination (**E, F**). Yeast expressing *R-yEmRFP* in the presence of Ubr1p expressed levels of fluorescence that did not depend upon illumination and were greater than that expressed by *PND-yEmRFP* (compare **G, H** to **C** and note the difference in exposure times), while fluorescence levels were highest when this construct was expressed in yeast lacking Ubr1p, regardless of illumination (**I, J**). These results indicate that the PND leads to blue light/Ubr1p dependent loss of yEmRFP activity. Moreover, in the context of yEmRFP, the presence of the phLOV2 domain in the PND results in a less stable protein than yEmRFP bearing a simple N-end rule-targeted arginine.

stronger fluorescence than did cells expressing PND-yEmRFP (note the differences in exposure time between Fig 2C and 2G). This suggests that unlike the situation observed for Ura3p, in which the simple amino terminal R-tagged protein version was apparently less stable than the PND-tagged version of Ura3p, the amino terminal R-tagged version of yEmRFP appeared to be more stable than the PND-tagged form.

**c. Cdc28p.** To investigate whether the blue light-induced loss of PND-containing proteins could rapidly produce a mutant phenotype such as cell cycle arrest, we generated *UBR1* and *ubr1Δ* strains bearing PND-tagged versions of the Cdc28p cell cycle regulatory protein. *CDC28* encodes a cyclin-dependent kinase that has multiple roles in the *S. cerevisiae* cell cycle [64]. Temperature-sensitive mutants of *cdc28* grown at non-permissive temperature [64,65], or cells in which the expression of a dominant negative version of Cdc28p have been expressed [66], arrest in the G1 phase of the cell cycle in an unbudded state. Cell growth continues, however, resulting in enlarged cells, including ones with long outgrowths, similar to the phenotype of cells exposed to mating pheromone [67]. For these experiments, we introduced the gene construct on the plasmid backbone derived from the yeast integrating plasmid, pPW66R [11]. After a homologous recombination event, the endogenous *CDC28* gene was interrupted by an insertion of the plasmid that also introduced a PND-HA-tagged version of *CDC28* under the transcriptional control of the *CUP1* copper-inducible promoter (Fig 3A).

*UBR1* cells expressing PND-Cdc28p that were plated in a single layer and incubated under illumination failed to generate colonies and were often arrested as individual cells (Fig 3C). Many of these cells exhibited the phenotype described above for TS mutants of *cdc28* grown at non-permissive temperatures [11,65] and for cells expressing the dominant-negative CDC28p [11,65,66]: enlarged cells, many with long outgrowths (Fig 3C). In contrast, when these cells were plated and incubated in the dark, they exhibited robust growth (Fig 3B), as did *ubr1Δ* cells expressing PND-HA-Cdc28p grown under both dark (Fig 3D) and light (Fig 3E) conditions. These cells exhibited normal size and morphology and many exhibited buds, consistent with normal growth and cell division. These results demonstrate that the PND tag directed blue-light dependent loss of Cdc28p activity that was rapid and sufficient to produce a cell cycle arrest phenotype.

## The PND directs light-dependent protein degradation

The results reported above strongly suggest that upon exposure to blue light, the PND facilitates Ubr1p-dependent ubiquitination of the fusion protein and its subsequent proteasomal degradation. To test directly whether the PND-dependent loss-of-function phenotypes were

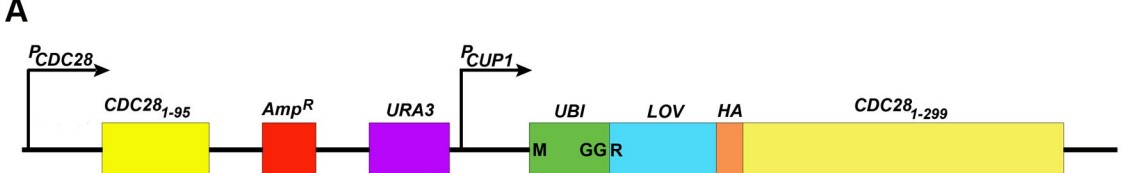

**A**

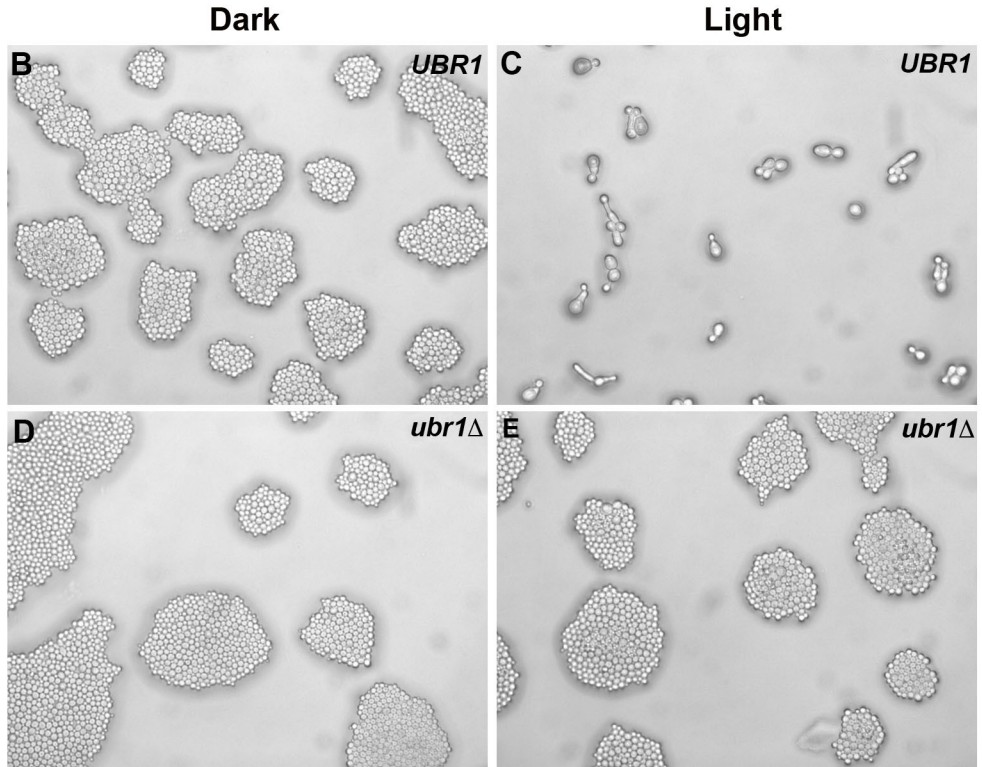

**Fig 3. Yeast cells in which the endogenous _CDC28_ gene has been replaced by _PND-HA-CDC28_ exhibit blue light/ Ubr1-dependent cell cycle defects.** A schematic diagram of the site of chromosomal insertion of the _PND-HA-CDC28_ transgene-bearing plasmid is shown in **(A)**. Homologous recombination results in the insertion of the entire plasmid at the genomic site of the restriction site (Msc I) that was used to linearize the plasmid. This results in the interruption of the endogenous _CDC28_ gene and its replacement by the PND-tagged form, under the transcriptional control of the copper-inducible CUP1 promoter. Cells were plated on selective medium and grown in the dark **(B, D)** and under blue-light illumination **(C, E)** in both _UBR1_ **(B, C)** and _ubr1Δ_ **(D, E)** genetic backgrounds. Note that UBR1 cells expressing PND-HA-Cdc28p under illumination **(C)** arrest as large single cells exhibiting long outgrowths, similar to what has been described for TS mutants of _cdc28_ grown at non-permissive temperatures [65], and for cells expressing the dominant-negative Cdc28p [66].

associated with protein loss, we carried out Western blot analysis of PND-HA-Cdc28p expressed under light or dark conditions. Starting with a fresh overnight culture grown in the dark, a small volume was inoculated into liquid selective medium and grown in darkness to early log phase (an optical density [OD] of approximately 0.2). At this point (T = 0), the culture was divided in half, with one culture continuing to grow in darkness while the other was grown under blue light illumination. Samples were taken at T = 0 and 5 subsequent hourly time points and processed for Western blotting.

As can be seen in Fig 4B, following exposure to light, the amount of PND-HA-Cdc28p in _UBR1_ cells decreased markedly in the first hour after exposure and remained at a low level for at least 5 hours. In contrast, when those cells were grown in darkness, the level of

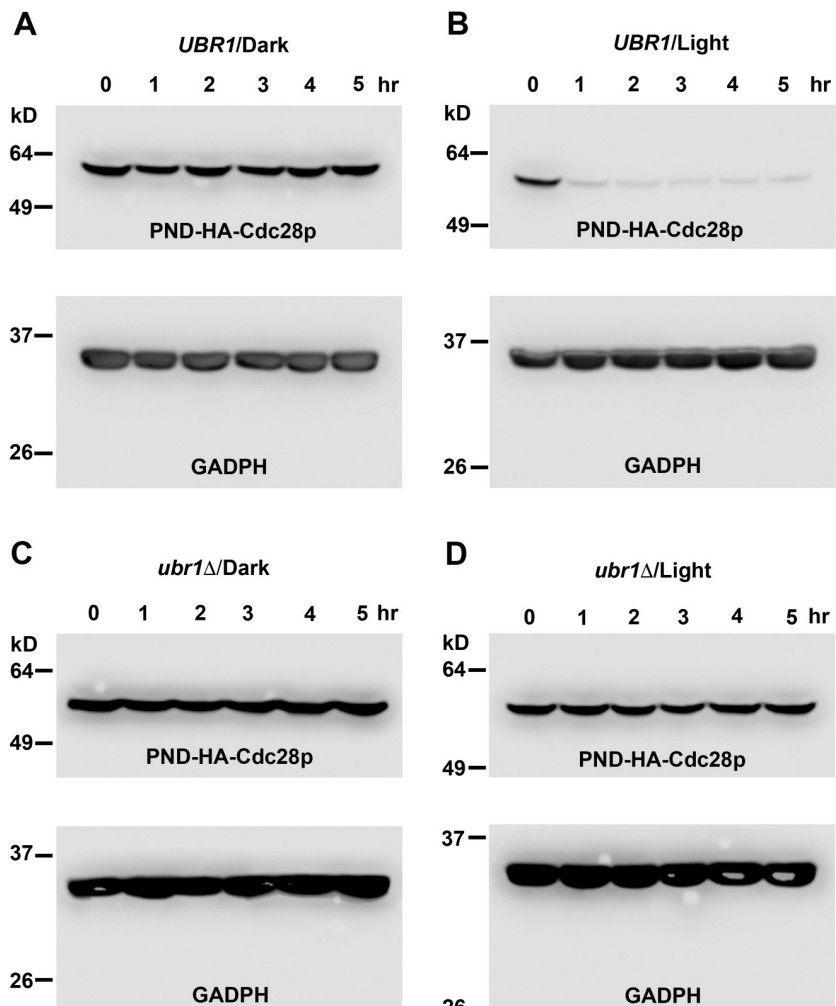

**Fig 4. PND-HA-Cdc28p undergoes blue light/Ubr1-dependent degradation.** Cells expressing a chromosomal insertion of PND-HA-Cdc28p in either a *UBR1* (**A, B**) or *ubr1Δ* (**C, D**) genetic background were grown in liquid culture in darkness to log phase, then divided and allowed to continue growth in darkness (**A, C**) or under blue-light illumination (**B, D**). Samples of culture medium were taken at 1-hour intervals and cells were processed for Western blot analysis. Western blots were divided into upper and lower sections with the upper sections probed using an antibody directed against the HA epitope in PND-HA-Cdc28p and the lower segments probed with an antibody directed against the endogenous protein Glyceraldehyde-3-phosphate dehydrogenase (GAPDH), which served as a loading control. Note that PND-HA-Cdc28p levels exhibited a dramatic decrease when grown in UBR1 cells under illumination (**B**). Growth in darkness (**A**) or under illumination in the absence of Ubr1p (**D**) resulted in constant levels of PND-HA-Cdc28p.

PND-HA-Cdc28p levels remained stable throughout the course of the experiment (Fig 4A). As expected, *ubr1Δ* cells expressed a steady level of PND-HA-Cdc28p when grown in the dark or under illumination (Fig 4C and 4D). Thus, the Ubr1p- and light-dependent loss of Cdc28p activity observed in *UBR1 ura3* cells was associated with a significant loss of PND-HA-Cdc28p, consistent with its light-dependent ubiquitination and degradation.

It has been shown for some proteins that the N-end rule degradation occurs post-translationally. For others, however, the presence of a destabilizing N-end, together with other protein-specific properties, leads to considerable degradation of nascent peptides in the process of translation (i.e. co-translational degradation) [68]. We realized that if the PND element were

primarily facilitating the degradation of nascent proteins during translation, its utility as a method for producing loss-of-function phenotypes would be considerably constrained. To investigate this possibility, we examined light-mediated loss of PND-HA-Cdc28p in *UBR1 ura3* cells grown in the presence and absence of the translational inhibitor cycloheximide. These experiments were carried out on a much shorter timescale than those described above, with cycloheximide added at T = 0 and samples taken every 15 minutes following the onset of illumination. Loss of PND-HA-Cdc28p from *UBR1 ura3* cells grown under illumination was rapid (Fig 5B), with most of the protein lost within 15 minutes after the onset of illumination. If the degradation of PND-HA-Cdc28p were occurring solely or primarily during translation, in cells in which translation was inhibited by cycloheximide there should be no marked difference in degradation rates seen in dark versus blue light conditions. In the presence of cycloheximide, there was still a rapid light-induced loss of PND-HA-Cdc28p (Fig 5D) that was not seen in dark conditions (Fig 5C), indicating that the degradation was not dependent upon concomitant translation. While this analysis does not rule out the possibility that some nascent PND-HA-Cdc28p undergoes Ubr1-mediated degradation during translation, it conclusively demonstrates that mature, full-length PND-HA-Cdc28p protein undergoes rapid degradation upon exposure to light, which allows the loss-of-function phenotype of *cdc28* to appear soon after the onset of illumination. The rapidity with which a PND-directed loss-of-function phenotype can be detected for a given protein, or indeed the rapidity with which the loss-of-function phenotype directed by any conditional degron can be detected, depends upon the rate of depletion of mature protein from the cells. Insofar as different proteins exhibit different intrinsic stabilities, this must be detected empirically for any protein-of-interest. Those proteins which exhibit both rapid degron-dependent co-translational degradation and rapid degradation of mature, synthesized protein are likely to be the best subjects for analysis using the PND as well as other conditional degrons.

Taken together, our analysis of the light-dependent loss of PND-HA-Cdc28p conclusively demonstrates that the PND represents a transferrable element that can confer rapid, blue light-dependent degradation of heterologous proteins via the N-end rule pathway, at least for some proteins. The rapid nature of PND-mediated degradation and the lack of significant levels of target protein perdurance are demonstrated by Western blot analysis and by our observations that *UBR1* cells expressing either PND-HA-Ura3p or PND-HA-Cdc28p under selective conditions often arrested as single cells under blue light illumination (Figs 1E and 3C), indicating that levels of protein required for function were depleted within one cell division cycle.

## The PND and the B-LID domain direct blue light-dependent protein loss-of-function and degradation in *Drosophila* embryos

Having demonstrated the effectiveness of the PND in eliciting light-dependent degradation in yeast, we were then interested to test the extent to which it could be used to generate light-dependent phenotypes in a multicellular organism. Accordingly, we examined the effects of the PND upon a modified version of the *Drosophila* dorsal-ventral (DV) patterning protein Cactus [69–71]. In early embryos produced by wild-type females, Cactus is distributed throughout the cytoplasm, where it binds to the Dorsal protein [69,71–73] and prevents it from entering the nucleus. Cactus undergoes graded ubiquitin/proteasome-dependent degradation along the DV axis in response to Toll receptor signaling on the ventral side of the embryo [74–76], thus releasing Dorsal to enter nuclei in a graded manner [77–79] with highest nuclear Dorsal levels on the ventral side of the embryo. Toll signaling and Cactus degradation occur over a brief time window during the syncytial blastoderm stage of embryogenesis, which

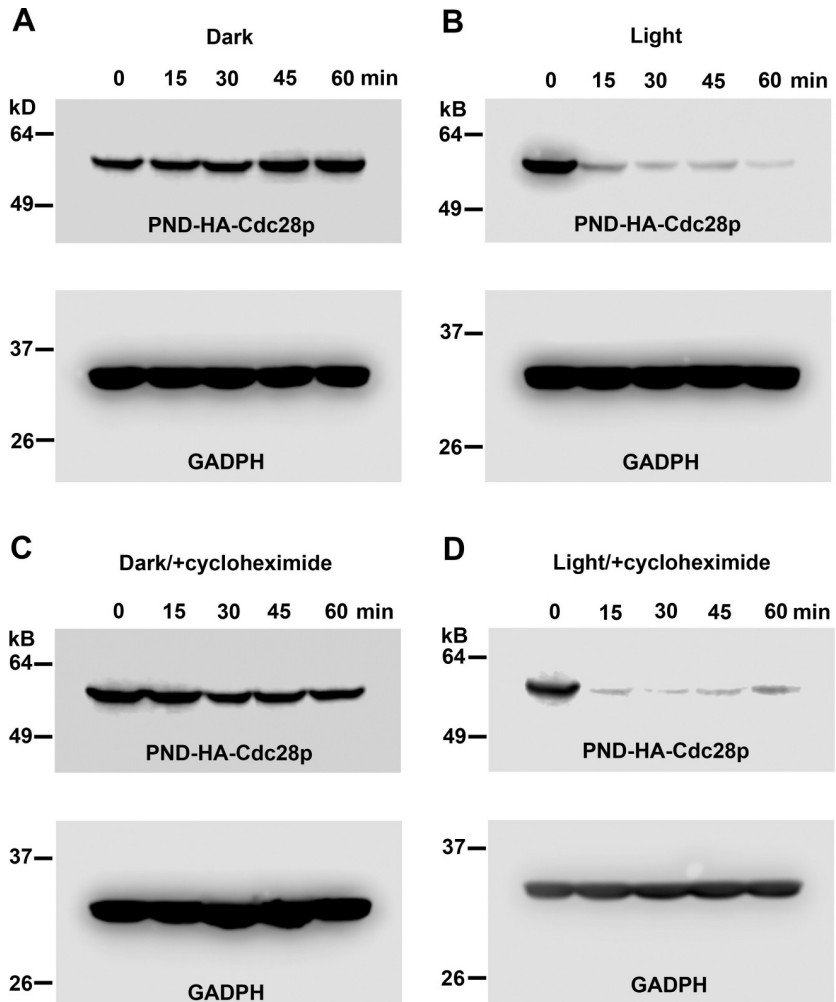

**Fig 5. Mature PND-HA-Cdc28P undergoes rapid light dependent degradation.** Cells expressing PND-HA-Cdc28p in a *UBR1* genetic background were grown in liquid culture in darkness to log phase, then divided and allowed to continue growth in darkness (**A, C**) or under blue-light illumination (**B, D**) in either the absence (**A, B**) or presence (**C, D**) of the translational inhibitor cycloheximide. Samples of culture medium were taken at 15-minute intervals and cells were processed for Western blot analysis, with an upper portion of each blot probed with an antibody against the HA epitope in PND-HA-Cdc28p and a lower portion probed with an antibody directed against endogenous GADPH, which served as a protein loading control. In the presence of Ubr1p, light-dependent loss of PND-HA-Cdc28p was very rapid, likely occurring within a single cell cycle, regardless of the absence (**B**) or presence (**D**) of cycloheximide.

makes Cactus an ideal candidate for testing the ability of the PND to elicit protein degradation and consequent loss-of-function phenotypes. We generated a transgene in which the PND-HA region was fused to the amino terminus of a modified version of Cactus [Cactus (3ala)] [80], in which serines 74, 78, and 116 have been converted to alanine residues (Fig 6A). Cactus(3ala) is insensitive to Toll receptor-dependent phosphorylation, ubiquitination and degradation. As a result, it binds constitutively to Dorsal protein, inhibits its nuclear uptake and is therefore dominantly dorsalizing. Accordingly, from this point we refer to Cactus(3ala) as CactDN (for Cactus Dominant Negative, owing to its dominant negative effect upon Dorsal function and DV patterning) and the PND-HA-tagged version as PND-HA-CactDN.

Several transgenic lines carrying PND-HA-CactDN (Fig 6A) were generated and the transgenes expressed under the control of the female germline-expressed Gal4 driver, *nanos*-Gal4:

## A  PND-HA-CactDN

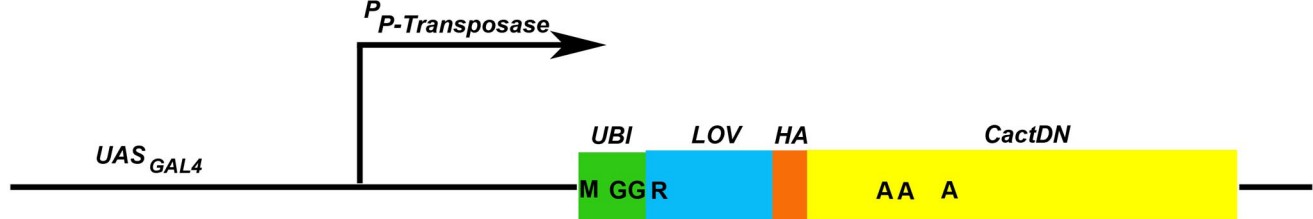

## B  CactDN-psd

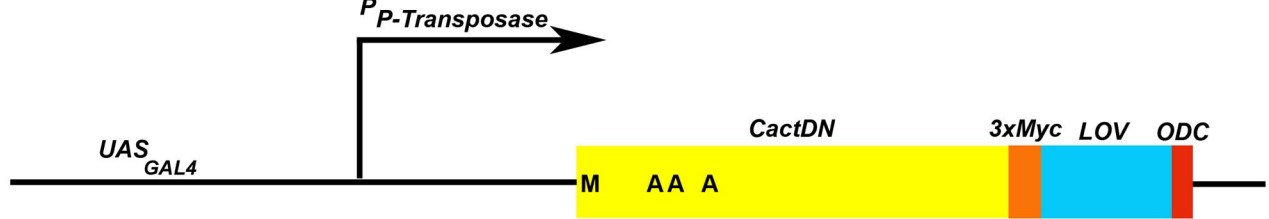

## C  CactDN-B-LID

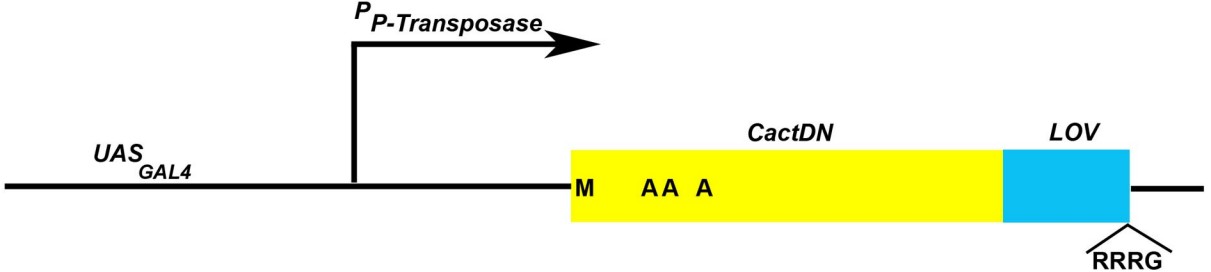

**Fig 6. Schematic diagrams of the transgene constructs encoding PND-, psd- and B-LID domain-tagged versions of the *CactDN* open reading frame.**
Constructs encoding the three degron-tagged versions of *CactDN* were introduced into the *Drosophila* genome on the P-element transposon-based expression vector, pUASp [110], downstream of upstream activator sequences for the yeast Gal4 transcription factor ($UAS_{GAL4}$) and the promoter from the P-element transposase gene ($P_{P-Transposase}$). Expression of the transgenes was accomplished by co-expression of a germline-specific source of the Gal4 transcription factor. **(A)** *PND-HA-CactDN*. **(B)** *CactDN-psd*. **(C)** *CactDN-B-LID*. Labels are as follows: *UBI*, a single copy of the ubiquitin open reading frame. *LOV*, encoding the LOV2 domain of plant phototropin I. *HA*, encoding a single copy of the influenza hemagglutinin (HA) epitope [61]. *3xMyc*, sequences encoding three tandem copies of the 9E10 epitope from human c-myc [113]. *ODC*, an element encoding 23 amino acids from the synthetic ODC-like degron [83]. The single letters A, G, M. and R, represent codons encoding individual alanine, glycine, methionine and arginine. Specifically, M's denote the initiation codons of the open reading frames of the three constructs. The three A's present in the *CactDN* segments represent 3 serine-encoding codons that were mutated to alanines, rendering the encoded protein insensitive to Toll pathway signal-dependent proteolysis. GGR in *PND-HA-CactDN* represents codons encoding the two glycine residues at the C-terminus of ubiquitin and the subsequent arginine residue at the N-terminus of the LOV element. Finally, RRRG represents the codons encoding the critical C-terminal residues of the B-LID domain, which are likely to support degradation by the DesCEND mechanism [100,101].

VP16 [81]. The hatch rates of embryos associated with 4 independent insertions of the PND-HA-CactDN-bearing transgene were pooled. Because CactDN is dominantly dorsalizing, we expected that dark-reared embryos derived from mothers carrying PND-HA-CactDN would have a very low hatch rate. Consistent with this prediction, all embryos failed to hatch.

In contrast, 90.4% of the embryos that were exposed to blue light starting within 1 hour of egg deposition and reared at 25°C hatched (Table 1). As noted above, the embryos cultured in darkness failed to hatch and these embryos were dorsalized (see below).

During the course of these studies, reports appeared which described the analysis of two other light-dependent degrons that rely upon the plant phototropin 1 LOV2 domain. Renicke et al. [56], engineered a photosensitive degron (psd), comprised of the phLOV2 from *Arabidopsis thaliana* combined with a synthetic peptide similar to the ubiquitin-independent degradation signal from murine Ornithine Decarboxylase (ODC) [82–84]. Proteins carrying the psd at their carboxy termini exhibited blue-light dependent degradation in yeast. Bonger et al. [26] showed that a four amino acid long peptide, arg-arg-arg-gly, when fused to the carboxy terminus of a protein-of-interest, led to rapid proteasome-mediated degradation in mammalian cells. This degron was combined with a modified version of the *Avena sativa* phLOV2 domain and showed that this blue light-inducible degradation (B-LID) domain could confer light-dependent degradation upon proteins in both cultured mammalian cells and zebrafish embryos [26,57]. To examine the effectiveness of these two degrons in *Drosophila*, we generated constructs and transgenic lines in which the psd and the B-LID domain were fused in-frame to the carboxy terminus of CactDN, referred to as CactDN-psd and CactDN-B-LID respectively (Fig 6B and 6C).

As was observed for PND-HA-CactDN, all of the CactDN-B-LID embryos that were cultured in the dark failed to hatch, while 84.8% of the embryos that were reared under blue light starting within 1 hour of egg deposition did hatch (Table 1).

When reared in darkness, none of the CactDN-psd embryos hatched. Similarly, when reared under the same blue light illumination conditions that had resulted in hatching PND-HA-CactDN and CactDN-B-LID embryos, no CactDN-psd hatchers were observed (Table 1).

Dorsalized embryos can be classed as falling into the following classes, based on the severity of the phenotype, which is determined based on the cuticular pattern elements present or absent as follows (Classifications are from Roth et al. [69], with modifications. See Fig 7 for representative phenotypes): completely dorsalized, lacking any dorsal/ventral polarity, D0; strongly dorsalized, D1; moderately dorsalized, D2; and weakly dorsalized, D3. The designation UH, seen in Fig 7 and Table 2, denote unhatched but otherwise, apparently normal embryos.

Both PND-HA-CactDN-expressing and CactDN-B-LID-expressing embryos exhibited dorsalized phenotypes when cultured in the dark. As noted above, when grown under illumination, most PND-HA-CactDN embryos hatched. In contrast, when they were grown in darkness the majority of these embryos exhibited either a moderate (D2) or weakly (D3) dorsalized cuticular phenotypes (Table 2). Despite some line-to-line variability, presumably owing to different levels of expression, for 8 of the 9 transgenic lines for which unhatched, dark-incubated embryos were counted and categorized, the largest cohort of embryos exhibited a D3 phenotype, followed by the cohort of embryos exhibiting a D2 phenotype. The small number

**Table 1. Hatch rates of light-exposed *Drosophila* embryos bearing the three degron-tagged versions of CactDN.** No embryos that were propagated in total darkness hatched.

| Maternal Genotype | Hatched Embryos | | Unhatched Embryos | | N |
| | % | SEM | % | SEM | |
|---|---|---|---|---|---|
| *UASp-PND-cactDN* | 90.4 | 0.2 | 9.6 | 0.2 | 1099 |
| *UASp-cactDN-B-LID* | 84.8 | 0.3 | 15.2 | 0.3 | 1485 |
| *UASp-cactDN-psd* | 0 | - | 100 | 0 | 513 |

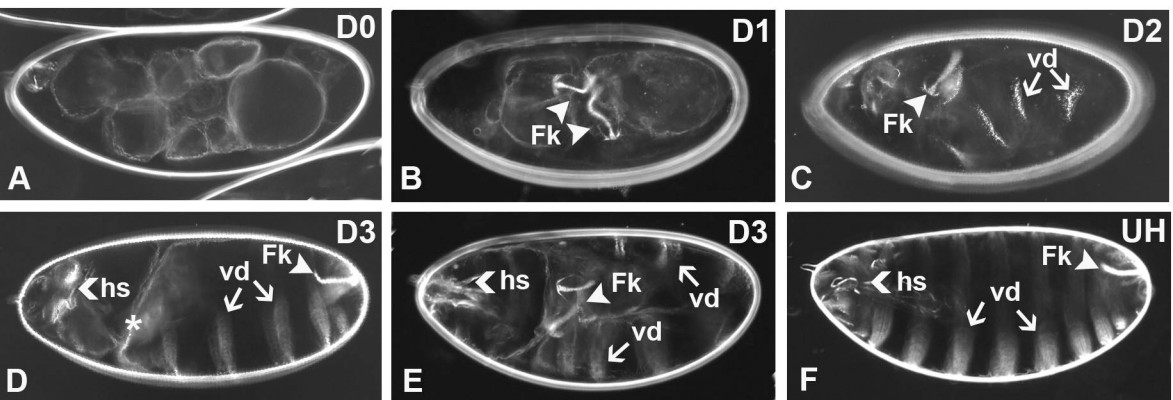

**Fig 7. Representative cuticular phenotypes of embryos expressing maternally provided degron-tagged *cactDN* constructs.** Embryos produced by females expressing the degron-tagged versions of CactDN described herein were collected and allowed to complete embryonic development in darkness, then subjected to cuticle preparation [112]. Levels of dorsalization denoted below are indicated at top right of each panel. **(A)** A completely dorsalized (D0) embryo produced by a female expressing *cactDN-psd*. **(B)** A strongly dorsalized (D1) embryo produced by a female expressing *cactDN-B-LID*. Note the presence of Filzkörper (Fk) structures (= tracheal spiracles). **(C)** A moderately dorsalized (D2) embryo from a female expressing *PND-HA-CactDN*. Note the presence of Filzkörper material and narrow ventral denticle (vd) bands. **(D)** A weakly dorsalized (D3) embryo from a female expressing *PND-HA-CactDN*, exhibiting the "twisted" phenotype. Note the asterisk marking the twist in the body axis. **(E)** A weakly dorsalized (D3) embryo, from a *PND-HA-CactDN*-expressing female, exhibiting the "U-shaped" or "tail-up" phenotype. **(F)** An apparently normal, unhatched (UH) embryo produced by a female expressing *PND-HA-cactDN*. In all panels, arrowheads mark the position of Filzkörper (Fk), arrows mark the position of ventral denticles (vd), and a left pointing angle mark (<) denotes the position of head skeletal (hs) elements. In all panels, anterior is to the left and the dorsal side of the egg is at top.

of light-exposed embryos that remained unhatched also included moderately and weakly dorsalized embryos (data not shown). Similarly, while most CactDN-B-LID embryos grown under illumination hatched, the embryos grown in darkness exhibited phenotypes ranging from completely dorsalized (D0) to weakly dorsalized (D3), with the largest number of embryos exhibiting a strongly dorsalized (D1) phenotype. This was the case both collectively and for the majority (6) of individual lines tested (9). In 2 of the 9 lines, D2 embryos were the largest cohort, while in one line, D0 embryos made up the largest cohort. Thus, despite the range in phenotypes among dark-grown embryos, CactDN-B-LID appears to be a more effective inhibitor of Dorsal protein function than PND-HA-CactDN; consequently, a lower proportion of illuminated CactDN-B-LID embryos hatch.

Because no CactDN-psd embryos exposed to light hatched, in order to determine whether light had any influence over the CactDN-psd protein, we compared the phenotypes of unhatched embryos grown under illumination with that of dark grown embryos (Table 2). In both cases, the majority of embryos exhibited a completely dorsalized D0 phenotype (in 18/18 transgenic lines tested). A small decrease in the proportion of D0 embryos and a small increase in the proportions of D1 and D2 embryos were observed in the embryos that were exposed to

**Table 2. Cuticular phenotypes of embryos bearing the three degron-tagged versions of CactDN, which had been propagated in total darkness.**

| Maternal Genotype | DO Embryos | | D1 Embryos | | D2 Embryos | | D3 Embryos | | UH Embryos | | *N* |
|---|---|---|---|---|---|---|---|---|---|---|---|
| | % | SEM | % | SEM | % | SEM | % | SEM | % | SEM | |
| *UASp-PND-cactDN* | 0.1 | 11.1 | 0.3 | 11.1 | 25.0 | 0.7 | 71.4 | 0.9 | 3.2 | 0.6 | 1217 |
| *UASp-cactDN-B-LID* | 13.7 | 2.2 | 54.4 | 0.9 | 27.6 | 1.0 | 4.3 | 1.2 | 0 | - | 1172 |
| *UASp-cactDN-psd* | 98.8 | 0.1 | 1.1 | 1.0 | 0.1 | 0.5 | 0 | - | 0 | - | 927 |
| *UASp-cactDN-psd* (with blue light) | 92.7 | 0.5 | 6.9 | 1.7 | 0.4 | 1.7 | 0 | - | 0 | - | 466 |

light (in 10/18 lines tested). However, if the trend observed for the effect of the three degrons upon CactDN were extended to other proteins-of-interest in *Drosophila*, the level of phenotypic changes elicited by the psd would be unlikely to be useful in phenotypic studies. However, as noted above in our studies of Ura3p and yEmRFP, bearing either an amino terminal PND or an amino terminal arginine residue, different degrons can elicit different levels of stability, in a protein dependent manner. Therefore, we cannot rule out the possibility for other proteins expressed in *Drosophila* embryos or other tissues, the psd may provide useful light-dependent changes in activity.

We also carried out Western blot analysis to assess the effect of blue light exposure upon protein levels of the PND-HA-CactDN and CactDN-B-LID transgenes and to examine how the embryonic phenotypes correlated with protein levels. Western blot analysis of extracts of embryos produced by PND-HA-CactDN- and CactDN-B-LID-expressing females was consistent with efficient light-dependent degradation of these two proteins (Fig 8). For each of these two constructs, extracts were generated from 2–4 hour-old embryos that had either been subjected to blue light illumination or allowed to develop in darkness. In order to avoid detection of the endogenous Cactus protein, Western blots of PND-HA-CactDN-expressing extracts were probed with an anti-HA antibody. Owing to the absence of the HA tag in CactDN-B-LID, an antibody directed against Cactus was used to probe blots bearing that fusion protein. Although the expected molecular weight of wild-type Cactus protein is 53.8 kD, it has been demonstrated that Cactus protein migrates on SDS-PAGE gels with an apparent molecular weight of 69–72 kD [72]; Developmental Studies Hybridoma Bank, University of Iowa). That, together with the addition of the PND or the B-LID domain was therefore expected to generate mature proteins of approximately 88–91 kD. Extracts from 2- to 4-hour old PND-HA-CactDN embryos that had been laid and incubated in the dark exhibited the presence of a band of approximately 90 kD, corresponding to PND-HA-CactDN, which disappeared in embryos incubated under illumination (Fig 8A). Similarly, extracts from 2- to 4-hour old CactDN-B-LID-expressing embryos from two independent transgenic lines exhibited a loss of the protein band corresponding to CactDN-B-LID in response to illumination (Fig 8B). Although the extent of protein loss differed between the two transgenic lines, presumably due to differences in expression between the two lines tested, in both cases a marked decrease in levels of CactDN-B-LID protein was detected in the extracts of light-exposed embryos, in comparison to their dark-incubated counterparts.

In order to more directly assess the phenotypic consequences of degron-mediated loss of CactDN activity, live imaging of embryos was carried out to visualize the behavior of fluorescent GFP-tagged Dorsal protein [85] expressed under the control of the endogenous *dorsal* gene transcriptional regulatory elements, together with each of the three degron-tagged versions of CactDN (Figs 9 and 10). Illumination of embryos with a blue laser (488nm) was performed to manipulate the degron-tagged proteins, enabling comparison of the dynamics of Dorsal nuclear accumulation controlled by PND-HA-CactDN, CactDN-B-LID, and CactDN-psd. In these experiments, the protein levels and activities were expected to vary depending on the length of exposure, the intensity of light, and the intrinsic stability of the degron fusion proteins, thus requiring optimization of the conditions of illumination. In this way, it was determined that embryos exposed to more than 20 min of high power 488nm wavelength light displayed developmental defects likely due to phototoxicity unrelated to effects upon DV patterning. Accordingly, in these experiments, embryos were first allowed to develop under low power (3.1%) 488nm laser illumination until early nuclear cycle (nc) 12. Embryos were then illuminated with blue light (488nm) for 20 minutes at 10% laser power (high power), a condition that permitted perturbation of CactusDN activity without eliciting phototoxicity. After 20 minutes of illumination, the embryos were returned to low power 488nm laser illumination in

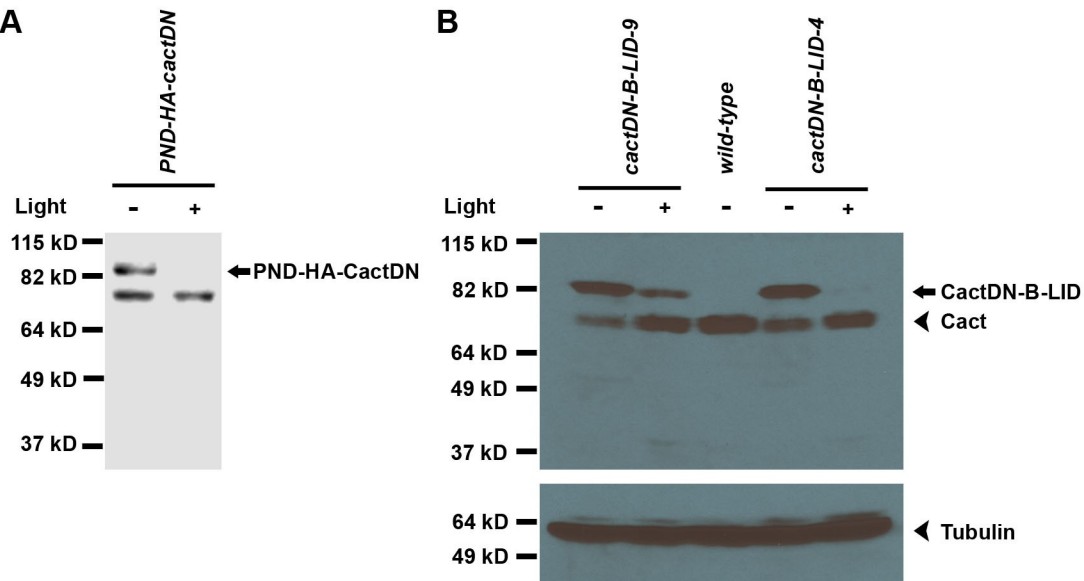

**Fig 8. PND-CactDN and CactDN-B-LID undergo light-dependent loss in *Drosophila* embryos.** Embryos from females expressing a transgene encoding PND-HA-CactDN (**A**) or from females expressing two independent transgenic insertions encoding CactDN-B-LID (**B**), were collected and allowed to develop in either darkness (-) or under blue light illumination (+). Embryonic extracts were prepared from 2–4 hour-old embryos and Western blots of those extracts were probed with antibodies directed against the HA epitope (**A**) and against Cactus protein (**B**) are shown. The positions of bands corresponding to PND-HA-CactDN, CactDN-B-LID, and endogenous Cactus (Cact) are shown.

order to limit further degradation of the degron-tagged CactDN proteins. In addition to activating the LOV domain chromophore associated with the light-dependent degrons, 488 nm light is also absorbed by and leads to emission by GFP. Nevertheless, 20 minutes of high power 488 nm laser light exposure did not result in Dorsal-GFP photobleaching that precluded its subsequent visualization.

Control embryos expressing Dorsal-GFP exhibited the formation of a normal Dorsal-to-Ventral nuclear gradient of the fusion protein (Fig 9B–9B'" and S1 Movie), even under blue-light illumination (Fig 9B' and 9B"). Prior to illumination at nc12, a point at which Dorsal-GFP had begun to enter the nuclei of the otherwise wild-type embryo (Fig 9B), embryos expressing each of the degron-tagged versions of CactusDN exhibited a perturbation of Dorsal-GFP nuclear uptake (Figs 9C, 9D, 9E, 9F, 10A and 10B). In PND-HA-CactDN- and CactDN-B-LID-expressing embryos that were exposed to low power 488nm laser light, Dorsal-GFP remained predominantly cytoplasmic through nuclear cycles 12–14 (Fig 9D–9D'" and S3 Movie, and Fig 9F–9F'" and S5 Movie), a phenotype which is explained by the continuing presence of degron-tagged CactDN protein; however, transient and sporadic low levels of nuclear Dorsal-GFP were observed at nc13 and nc14, likely owing to a slow rate of degradation of degron-tagged CactDN occurring in the presence of low intensity blue light. These low levels of nuclear Dorsal-GFP are consistent with the dorsalized cuticular phenotypes observed for most dark-cultured PND-HA-CactDN and CactDN-B-LID embryos (Table 2). In contrast to their low power-illuminated counterparts, PND-HA-CactDN- and CactDN-B-LID-expressing embryos that were exposed to high power blue laser light exhibited nuclear accumulation of Dorsal-GFP during nc13 (Fig 9C' and S2 Movie, and Fig 9E' and S4 Movie, respectively) and by nc14, these embryos exhibited conspicuous ventral-to-dorsal nuclear gradients of Dorsal-GFP (Fig 9C" and S2 Movie and Fig 9E" and S4 Movie). By stage 6 of embryogenesis, ventral cells within these embryos began to display normally polarized cell movements (Fig 9C'" and

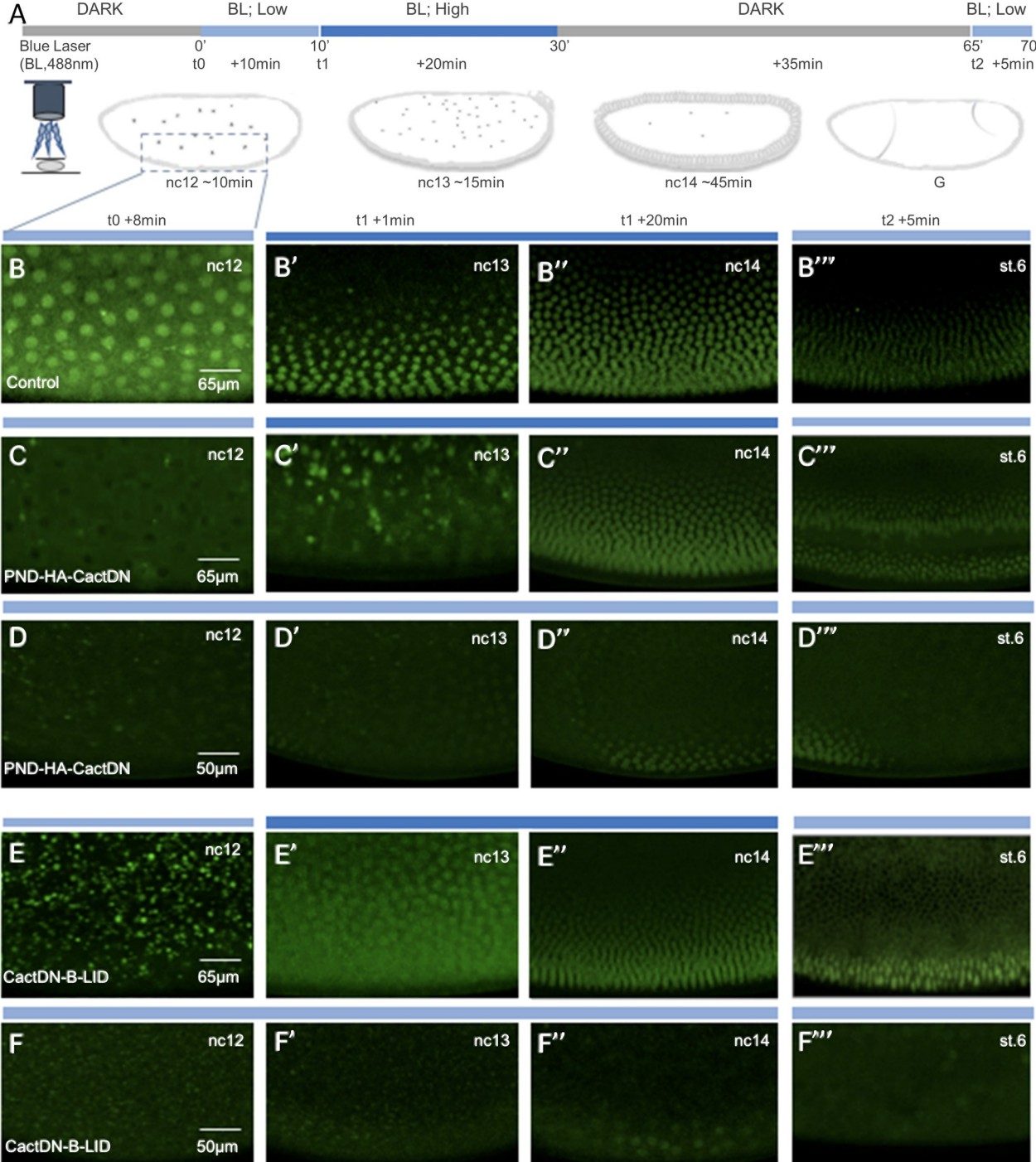

**Fig 9. Laser illumination of live embryos expressing PND-HA-CactDN or CactDN-B-LID induces nuclear accumulation of Dorsal-GFP. (A)**
Schematic showing the imaging setup that was used to visualize Dorsal-GFP in *Drosophila* embryos over a period of ~75 min spanning their development from nuclear cycle (nc) 12 up to gastrulation (st.6) under conditions that inactivate Cactus-degron fusions. Imaging was initiated at time = 0 (t0) and continued for a period of ~10 minutes during nc12 under low power 488 nm laser illumination. Immediately after this treatment (at t1) and extending into nc13 (a period of 15 minutes), embryos were illuminated for 20min under high power 488nm laser to initiate degron-mediated loss of CactusDN. After a 30–35' rest in the dark at which point embryos had initiated gastrulation (t2), they were again illuminated for 5 min under low power 488nm laser to monitor the Dorsal-GFP gradient and the developmental state of the embryos (t2 + 5min). The dotted box represents the illuminated area. The remainder of the panels show four snapshots each, taken from movies of embryos containing Dorsal-GFP [85], either expressed alone (**B-B‴**, control; see also S1 Movie) or together with the PND- and B-LID-tagged Cactus variants expressed under the control of the *mat-α4-tub-Gal4*:*VP16* driver element [108]. The PND-HA-CactDN (**C-C‴**, **D-D‴**; see also S2 and S3 Movies) and the

CactDN-B-LID (**E-E'''**, **F-F'''**; see also S4 and S5 Movies) fusion proteins were imaged using conditions outlined in panel **A** (**C-C'''**, **E-E'''**, S2 and S4 movies) or under low power 488nm laser illumination (light blue bar)(**D-D'''**, **F-F'''**, S3 and S5 Movies). Scale bars represent 65μm or 50μm, as noted; in the absence of Dorsal-GFP nuclear translocation, we used a slightly higher digital magnification (i.e. 50μm), in those cases to increase visibility of empty nuclei.

9E'''), consistent with the onset of ventral furrow formation. The normal polarization of nuclear Dorsal-GFP accumulation and cell movements is presumably due to the loss of degron-tagged CactDN protein, enabling endogenous wild-type Cactus protein to engage with and control Toll receptor signal-mediated nuclear uptake of the Dorsal-GFP fusion protein. Together, the comparable nature of phenotypes observed via confocal microscopy coupled with laser illumination, and by cuticle preparations following overhead blue-light illumination with a grid of LED bulbs strongly supports the use of the PND and the B-LID domain as effective tools for controlled elimination of targeted proteins-of-interest in *Drosophila* embryos. Moreover, the observation of substantial nuclear accumulation of Dorsal-GFP as early as 1 minute after high power blue light illumination (see Fig 9E' and S4 Movie, which was obtained 1 minute after the onset of high-power blue laser light illumination) demonstrates the utility of these elements for the analyses of loss-of-function phenotypes requiring fine time resolution and/or rapid onset.

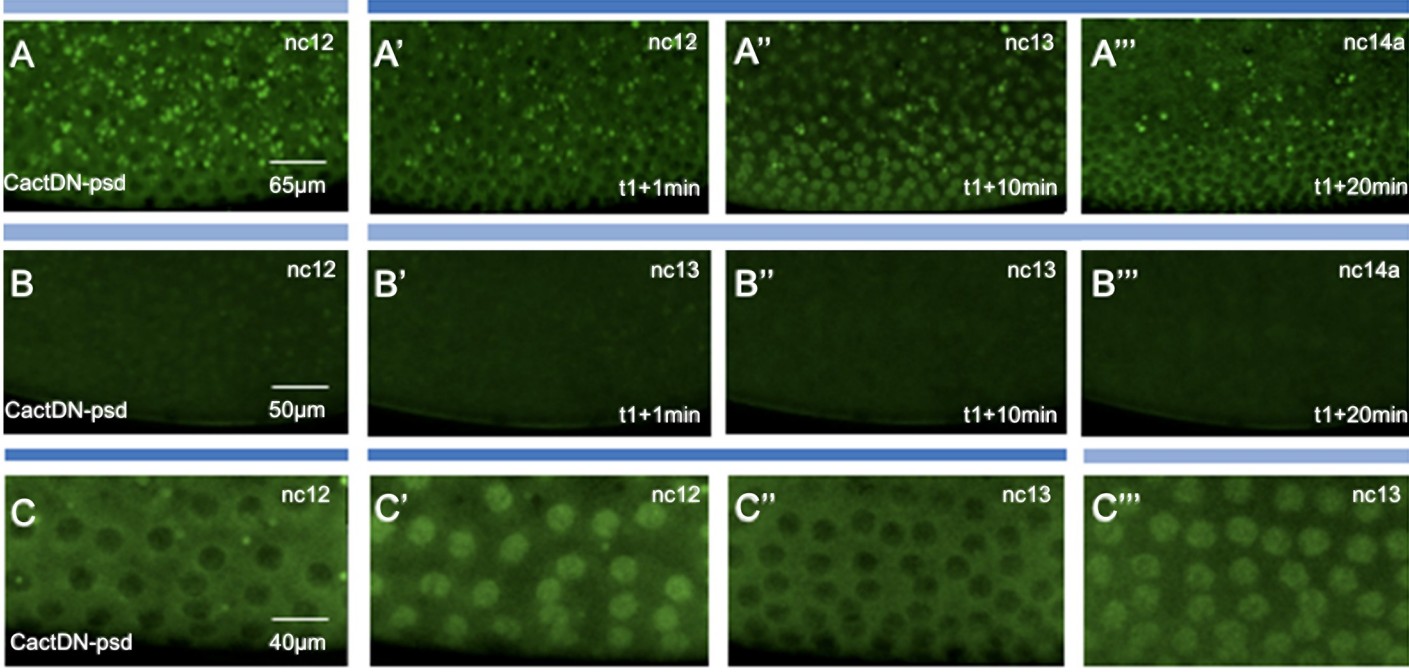

**Fig 10. Laser illumination of live embryos expressing CactDN-psd induces transient cyclical nuclear accumulation of Dorsal-GFP.** Images shown are four snapshots taken from movies of embryos expressing Dorsal-GFP, together with the photosensitive degron-tagged CactusDN (CactDN-psd) expressed under the control of the *mat-α4-tub-Gal4:VP16* driver element, imaged under different conditions. Panels represent snapshots from respective S6–S8 Movies. (**A-A'''**, **B-B'''**, **C-C'''**) Imaging was initiated at time = 0 (t0) and continued for 10 minutes during nc12 under low power 488 nm, using the scheme diagrammed in Fig 9A. Just after this treatment (i.e. t1) and extending into nc13 (t1+15min) and nc14a (t1+20min), embryos were illuminated for 20min at 488nm high power to initiate degron-mediated loss of CactDN-psd (**A'-A'''**; see also S6 Movie). As a control, embryos were also imaged under low power 488nm only (**B-B'''**; see also S7 Movie). Scale bars represent 65μm or 50μm, as noted; in the absence of Dorsal-GFP nuclear translocation, we used a slightly higher digital magnification (i.e. 50μm), to increase visibility of empty nuclei. (**C-C'''**) Embryos were exposed to blue light earlier for 20 min, initiating at nc12 and into nc13 (blue bar, 488nm), and subsequently imaged under low power 488nm laser illumination. These images show that Dorsal-GFP enters nuclei in a transient manner, entering just before division but relocalize to the cytoplasm after nuclear division; see also S8 Movie.

As noted above, in CactDN-psd-expressing embryos that were not exposed to high power blue light, Dorsal-GFP protein was never detected predominantly in nuclei (Fig 10B–10B'" and S7 Movie), consistent with the completely dorsalized cuticular phenotypes exhibited by CactusDN-psd cultured in darkness (Fig 7A). Dorsal-GFP was also present predominantly in the cytoplasm of illuminated CactDN-psd embryos (Fig 10A, 10A', and 10A'" and S6 Movie, and Fig 10C and 10C" and S8 Movie) consistent with the cuticular phenotypes and with a greater stability, lower sensitivity to blue light, and/or slower rate of degradation than either PND-HA-CactDN, or CactDN-B-LID. However, these embryos did exhibit a brief cell cycle-dependent period of Dorsal-GFP nuclear localization of about 1–2 minutes immediately prior to the mitoses of nuclear cycles 13 and 14 (Fig 10A", 10C', and 10C'", and S6 and S8 Movies). These results may indicate that in the embryo, where the relatively stable CactDN-psd is continuously being translated from maternally provisioned mRNA, the high intensity blue light provided by a laser is sufficient to eliminate enough CactDN-psd by the end of a nuclear cycle to allow Dorsal-nuclear uptake on the ventral side of the embryo, with continued synthesis of CactDN-psd following mitosis again being sufficient to sequester Dorsal-GFP in the cytoplasm. Alternatively, these particular conditions may reveal a previously unappreciated cell cycle-dependent enhancement of either Dorsal nuclear uptake, or of psd-mediated proteasomal degradation immediately prior to mitosis in early *Drosophila* embryos. A conclusive explanation of these events requires the development of a fluorescently-tagged version of CactDN-psd that would permit direct live imaging of the behavior of this protein in response to blue laser light.

Based on the observations reported above, both the PND and the B-LID domain confer easily distinguished light-dependent phenotypes when fused to CactDN and therefore exhibit promise for use in the analysis of phenotypes associated with loss-of-function for other proteins, at least in the context of the early embryo. In view of the current discrepancy in phenotypes elicited in CactDN-psd in response to incident versus laser illumination, we cannot currently conclude that the psd element is a generally useful tool for studies aiming at perturbing the action of tagged proteins-of-interest in *Drosophila* embryos. However, when fused to other proteins, expressed in other tissues, or under different treatment regimens, the psd might direct useful, light-dependent changes in protein levels and function.

## Discussion

Our studies clearly demonstrate that the Photo-N-degron can be a valuable tool for the generation of conditional loss-of-function phenotypes in yeast. Previous reports describing the psd [56] and the B-LID domain [56,57] provided us with the opportunity to compare the effectiveness of the obligately N-terminal PND with that of the obligately C-terminal psd and the B-LID domain, in mediating light-dependent degradation of our model protein Cactus in *Drosophila* embryos. Insofar as the addition of an N- or C-terminal extension can disrupt the function of some proteins, the availability of light-dependent degrons that can function at either the N- or C-termini of proteins increases the versatility and likelihood of success for investigations involving light-mediated protein degradation. Our analyses indicate that the PND that we developed, as well as the B-LID domain, but not the psd, are capable of simply and effectively mediating temporally-specific elimination of CactDN within the single cell syncytial blastoderm embryo.

One advantage of light-induced degrons is that they can act rapidly and allow much greater temporal control than degrons regulated by the application or depletion of small molecules or by the induced expression of protein activators. Moreover, exposure to blue light is unlikely to produce the changes in enzyme or cellular behavior that changes in temperature are likely to

engender. However, our attempts to determine the extent to which the light-dependent degrons can mediate precise spatially-restricted protein degradation using CactDN as a target have been inconclusive. Further studies, likely involving other protein targets, are necessary to resolve this issue. However, subcellular resolution employing lasers to provide illumination [39,40] has been demonstrated with a variety of other optogenetic techniques, so there is reason for optimism that light-dependent degrons may also be able to provide fine spatial resolution of protein degradation.

A potential drawback to the light-inducible degron approach is that the target cells/tissues must be accessible to light. In our experiments using PND-tagged Ura3p and Cdc28p in yeast cells grown on agar plates for example, strong loss-of-function phenotypes were only detected when cells were distributed in a single layer and illuminated. Light-dependent degrons have been shown to be effective for phenotypic analysis in single cells [56,57] and in transparent organisms such as the nematode *Caenorhabditis elegans* [86] and the zebrafish [57]. In addition to yeast cells, our studies indicate that *Drosophila* eggs/embryos are sufficiently transparent to enable phenotypic studies of the effects of protein loss using light-dependent degrons. The use of light-dependent degrons in organisms that are not transparent is likely to be more technically challenging. However, optogenetic studies involving light-activated ion channels have been carried out in mice using fiberoptic delivery of blue light or small wirelessly powered light-emitting implants [87–90], suggesting this approach might be applied to the use of light-activated degrons.

In our studies of degron-tagged CactDN, the fusion proteins were not supplying normal Cactus function but rather were exerting dominant negative inhibition on Dorsal nuclear localization leading to embryo dorsalization. For most applications, we envision a more conventional use of these degrons that would involve the introduction of a functional degron-tagged version of protein-of-interest substituting for the corresponding endogenous gene, similar to our analysis of degron-tagged versions of Ura3p and Cc28p in yeast. In this context, blue light-mediated degradation of the degron-tagged protein would reveal the loss-of-function phenotype of the protein-of-interest. This would require that the degron-tagged version of the protein retain sufficient functional activity to be able to rescue the mutant phenotype to viability in the dark and to exhibit sufficient sensitivity to light for function to be eliminated or strongly diminished under illumination.

A variety of factors should be considered when utilizing light-dependent or other classes of conditional degrons in an experimental context. As noted above, the first consideration is the position at which the degron will be placed in the protein-of-interest. Currently, possible locations are limited to the N- or C-termini of proteins-of-interest. However, degrons that can be introduced at internal sites within proteins may be developed in the future. Secondly, individual proteins can exhibit widely different intrinsic stabilities, with mammalian proteins exhibiting half-lives ranging from 10 minutes to over a century [91,92], and they can behave differently when tagged with particular degrons. Our studies of CactDN in *Drosophila* embryos show that different degrons can influence the stability of the same protein to different extents, even in the uninduced state. While CactDN-psd led to complete dorsalization of expressing embryos under dark conditions, indicating that the degron-tagged protein is very stable, PND-HA-CactDN did not. Under dark conditions, most embryos expressing this construct exhibited only weak or moderate dorsalization. It is likely that the presence of an N-end arginine residue destabilizes CactDN to some extent, even under dark conditions. Similarly, only 13.7% of embryos expressing CactDN-B-LID under dark conditions produced completely dorsalized DO embryos, indicating some B-LID domain-dependent degradation of the fusion protein even in the dark. In addition to the choice of degron used in a particular study, these observations also have important implications for how degron-tagged proteins should be

expressed in experimental investigations. The advent of CRISPR/Cas9-directed approaches for gene replacement [93,94] provides the opportunity to simultaneously eliminate endogenous gene (and protein) expression while placing the degron-tagged version of the gene under correct spatial and temporal control of transcription. Moreover, the availability of these CRISPR/Cas9-dependent gene replacement approaches make possible similar conditional phenotypic studies using degron-tagged proteins in non-traditional model organisms [95–98]. However, as the addition of a degron to the protein-of-interest is likely to reduce the stability of that protein in comparison to its wild-type counterpart even in the absence of illumination, the expression levels of the degron-tagged protein by the CRISPR/Cas9-introduced gene may be insufficient to provide wild-type levels of rescuing protein. In such cases, an alternate approach for expression may be necessary, in which the rescuing degron-tagged protein is transcribed at higher than endogenous levels, in a genetic background homozygous for loss-of-function alleles of the corresponding endogenous gene. In support of this possibility, our recent study found that while Dorsal-B-LID fusions exhibited effective photosensitive degradation [99], the expression levels of the CRISPR-introduced construct were lower than that of the endogenous protein with embryonic phenotypes suggesting that some degradation of the protein was occurring in the absence of illumination.

It is not yet clear whether the behavior of CactDN, under the influence of the three light-dependent degrons reflect differences in the intrinsic stabilities of the fusion proteins, or differences in the rates and/or effectiveness of the ubiquitin/proteasomal pathways targeting their degradation, or both. The three degrons present in these constructs are targeted by different pathways leading to proteasomal degradation. PND-directed protein degradation operates via the Ubr/N-recognin class of ubiquitin E3 ligases [15–17]. The structure of the B-LID domain suggests that its degradation is mediated by the DesCEND (Destruction via C-end degrons) mechanism, via Cul2, a RING domain-containing ubiquitin ligase, together with an Elongin B/C protein [26,100,101]. The psd is similar to the destabilizing element present in mouse ODC [82–84], a very labile protein that undergoes proteasome-mediated degradation that is independent of ubiquitin. This lability likely results from its lack of a stable structure, together with the presence of a cysteine-alanine rich domain that is involved in recognition by the proteasome [102–104]. More recently, however, it has been shown that the light-activated degradation of some psd-tagged ER transmembrane proteins as well as some psd-tagged soluble cytoplasmic proteins is dependent upon the ERAD-C and its associated ubiquitination machinery [105,106]. Accordingly, experimental utilization of any individual degron will require the presence of the relevant degradation machinery in the cells in which the protein-of-interest is to be targeted for elimination.

Based on our studies, experiments designed to eliminate proteins-of-interest in early *Drosophila* embryos would be more likely to be informative if either the B-LID domain or the PND were utilized, at least under the overhead LED illumination conditions used for embryos on plates. Though not effective in eliciting light-dependent degradation of CactDN in embryos, the extent to which the psd can elicit light-dependent degradation of other proteins, or in other tissues in *Drosophila*, remains to be determined. Effective degradation of some proteins under the control of the psd might require more intense blue light illumination provided by a laser. Alternatively, for proteins that are extremely labile, the psd might be optimal for the detection of robust differences in phenotype that are dependent upon blue light. It is worth pointing out that although the psd was not effective in eliciting the degradation of CactDN in *Drosophila* embryos illuminated on plates, in *C. elegans*, light-induced degradation of psd-tagged Synaptotagmin resulted in a robust reduction of locomotion within 15 minutes of illumination [86]; within one hour, worm behavior and patch-clamp recordings of miniature postsynaptic currents were affected almost to the same degree as observed in worm mutants

for *snt-1*, a loss-of-function mutation in the gene encoding Synaptotagmin. As should be clear from the discussion above, a significant challenge to the application of inducible degron technology to the analysis of protein function is identifying an appropriate degron system that is "tuned" to the particular protein under study. Thus, the most effective degrons for different proteins and different tissues will vary and some initial empirical analysis will likely be required in identifying the right degron for the job.

## Materials and methods

### Yeast and *Drosophila* strains and maintenance

Yeast strains YPH500 (*MATalpha ade2-101 his3-Δ200 leu2-Δ1 lys2-801 trp1-Δ62 ura3-52*) and JD15 (*MATalpha ade2-101 his3-Δ200 leu2-Δ1 lys2-801 trp1-Δ62 ubr1Δ::LEU2 ura3-52*) were kind gifts of Dr. Jürgen Dohmen. Yeast were grown in liquid culture in either YPD or Synthetic Complete drop-out (SD) media with added supplements, but lacking those amino acids necessary for selection of introduced plasmid elements, or for selection for URA3 activity. Growth on plates was on YPD or SD media supplemented with 2% Bacto-agar. For the induction of genes under the control of the *CUP1* promoter, Cupric sulfate was added to medium at a concentration of 0.1 mM using a 1000x stock solution (0.1M Cupric sulfate in water). Plasmids were introduced into yeast using the LiAc/SS carrier DNA/PEG method [107].

The wild-type *Drosophila melanogaster* strain used to generate transformant lines in this study was a $w^{1118}/w^{1118}$ mutant derivative of OregonR. Fly stocks were maintained and embryos collected employing standard conditions and procedures. The *nanos-Gal4*:*VP16* transcriptional driver element is described in Van Doren et al. [81]. The *mat-α4-tub-Gal4*:*VP16* transcriptional driver element is described in Häcker and Perrimon [108]. The 25 kb transgene encoding *dorsal-GFP* under the control of the endogenous *dorsal* gene transcriptional elements is described in Reeves et al. [85].

### Plasmid constructs

**Yeast.** Plasmids pPW43 and pPW17R (both kind gifts of Dr. Jürgen Dohmen) are yeast centromere plasmids derived from plasmid pNKY48 [109], which bear sequences encoding the UBI-R-DHFR^ts N-degron, and its wild-type counterpart (UBI-R-DHFR), fused in frame to one copy of the HA epitope [61], followed by the Ura3p open reading frame. Transcription of the gene expressing the Ura3p fusion protein is driven by the copper-inducible *CUP1* promoter. These plasmids served as the starting point for the engineering of constructs bearing the light-sensitive degron described in this work. Initially, both plasmids were restriction digested with NotI, and the cut ends filled in, followed by recircularization, resulting in the elimination of the unique NotI site in both plasmids, thus yielding *pPW43-NotI⁻* and *pPW17R-NotI⁻*.

Both of these plasmids were then subjected to mutagenesis by inverse PCR, followed by recircularization by ligation, using the following two oligonucleotides: /5'Phos/ TCCGTGGCGGCCGCCTCTTAGCCTTAGCACAAGATGTAAG and /5'Phos/ TCCGGCGGGCGCGCCATGGTTCGACCATTGAACTGCATCG, yielding the two plasmids *pPW-UBI-R-NotI/AscI-DHFR^ts-HA-URA3* and *pPW-UBI-R-NotI/AscI-DHFR-HA-URA3*. This step placed the last two glycine codons of yeast ubiquitin and the subsequent arginine codon in the context of a NotI site and placed an additional AscI site between the NotI site and the sequences encoding DHFR^ts and DHFR, respectively. *pPW17R-NotI⁻* was also subjected to mutagenesis by inverse PCR, followed by recircularization by ligation, using the following two oligonucleotides: /5'Phos/TCCGTGGCGGCCGCCTCTTAGCCTTAGCACAAGATGTAAG

and /5'Phos/TCCGGCGGGCGCGCCGGTACCTACCCA, yielding the plasmid *pPW-UBI-R-NotI/AscI-HA-URA3*, from which the DHFR sequences had been excised.

The two oligonucleotides: 5'GGCTAAGAGGCGGCCGCTTGGCTACTACACTTGAACGTATTGAG and 5'CGAACCATGGCGCGCCCAAGTTCTTTTGCCGCCTCATCAATATTTTC were used for high fidelity PCR amplification of a DNA fragment encoding 143 amino acid long segment of the *A. sativa* phototropin 1 protein, encompassing the LOV2 domain, using a DNA clone encoding the *phot1* gene (a kind gift of Drs. Tong-Seung Tseng and Winslow Briggs) as a template.

Similarly, the two oligonucleotides: 5' GGCTAAGAGGCGGCCGCAGCCATACCGTGAACTCGAGCACCATG and 5'CGAACCATGGCGCGCCCTTCCGTTTCGCACTGGAAACCCATGCTG were used for high fidelity PCR amplification of a DNA fragment encoding the 185 amino acid long *Neurospora crassa* Vivid protein [50] minus its initiation codon, but including its associated LOV domain (vvdFL), using a *vivid* cDNA (a kind gift of Drs. Arko DasGupta, Jay Dunlap and Jennifer Loros) as a template.

Similarly, the two oligonucleotides: 5'GGCTAAGAGGCGGCCGCCATACGCTCTACGCTCCCGGCGGTTATGAC and 5'CGAACCATGGCGCGCCCTTCCGTTTCGCACTGGAAACCCATGCTG were used for high fidelity PCR amplification of a DNA fragment encoding a 150 amino acid long stretch of *Neurospora crassa* Vivid protein lacking the first 36 codons, which corresponds to the region encoding the LOV domain (vvdLOV), using a *vivid* cDNA (a kind gift of Drs. Arko DasGupta, Jay Dunlap and Jennifer Loros) as a template.

The resulting amplification products were digested with NotI and AscI and ligated to similarly digested *pPW-UBI-R-NotI/AscI-DHFR^{ts}-HA-URA3*, *pPW-UBI-R-NotI/AscI-DHFR--HA-URA3*, and *pPW-UBI-R-NotI/AscI-HA-URA3*, yielding the following plasmids:

- *pPW-UBI-R-phLOV2-DHFR^{ts}-HA-URA3*

- *pPW-UBI-R-vvdFL*(for full-length*)-DHFR^{ts}-HA-URA3*

- *pPW-UBI-R-vvdLOV-DHFR^{ts}-HA-URA3*

- *pPW-UBI-R-phLOV2-* DHFR-*HA-URA3*

- *pPW-UBI-R-vvdFl-DHFR-HA-URA3*

- *pPW-UBI-R-vvdLOV-DHFR-HA-URA3*

- *pPW-UBI-R-phLOV2-HA-URA3 (= pPW-PND-HA-Ura3p)*

- *pPW-UBI-R-vvdFL-HA-URA3*

- *pPW-UBI-R-vvdFLOV-HA-URA3*

yEmRFP is a yeast codon optimized version of the mCherry mRFP variant [63]. To generate a construct that expresses a PND-tagged version of yEmRFP the two oligonucleotides: 5'GTCACTGAGGCGCGCCATGGTTTCAAAAGGTGAAGAAGATAATATGGC and 5'GGGTTATTTATATAATTCATCCATACCACCAG were used for high fidelity PCR amplification, using a cDNA encoding yEmRFP (a kind gift of D. Neta Dean) as a template. The resulting amplification product was digested with AscI and ligated to AscI/SmaI-digested *pPW-UBI-R-phLOV2-HA-URA3*. This resulted in the replacement of sequences encoding the HA tag and URA3 by the yEmRFP open reading frame, fused in frame to the PND, in plasmid *pPW-PND-yEmRFP*. The same AscI/SmaI digested PCR fragment was also ligated to AscI/SmaI-digested *pPW-UBI-R-NotI/AscI-HA-URA3*, again replacing the HA-tag and URA3 by the yEmRFP open reading frame, in this case generating an yEmRFP construct bearing an N-

end rule targeted amino terminal arginine residue, but which does not contain the phLOV2 element (*pPW-R-yEmRFP*).

Plasmid *pPW66R* is a yeast integrating vector in which the sequences encoding the DHFR<sup>ts</sup> N-degron have been placed upstream and in-frame with the sequences encoding the HA epitope tag and the first 95 codons of the *CDC28* gene [11]. Recombination between the *CDC28* sequences present on the plasmid and the endogenous chromosomal *CDC28* gene results in the interruption of the endogenous *CDC28* gene and its replacement by an *DHFR<sup>ts</sup> N-degron-* and *HA* epitope-tagged version of the full-length 299 codon *CDC28* gene, under the control of the *CUP1* promoter. This recombination event also results in the insertion of a wild-type version of the *URA3* gene, permitting selection for chromosomal insertions on medium lacking uracil. *pPW-UBI-R-phLOV2-HA-URA3*(*pPW-PND-HA-Ura3p*) contains an AgeI/KpnI restriction fragment that includes the sequences encoding most of the ubiquitin protein and the entire LOV domain of the PND. Substitution of this AgeI/KpnI fragment for a corresponding fragment in *pPW66R* results in a precise replacement of the DHFR<sup>ts</sup> N-degron by the PND. However, prior to carrying out this subcloning step, it was first necessary to eliminate a second KpnI site present in *pPW66R*. Accordingly, we carried out partial digestion of *pPW66R* with KpnI in the present of ethidium bromide. Linear DNA obtained following this digestion was treated with Klenow enzyme in the presence of dNTPs to generate blunt ends, then recircularized by ligation, followed by screening for clones in which the correct KpnI site had been destroyed (*pPW66R-1Kpn*). The *AgeI/KpnI* DNA restriction fragment encoding the PND was isolated from *pPW-PND-HA-Ura3p* and ligated to AgeI/KpnI-digested *pPW66R-1Kpn* from which the *DHFR<sup>ts</sup>* N-degron sequences had been excised, resulting in plasmid *pPW66R-PND-HA-cdc28*. This plasmid was linearized with MscI, which digests a site within the *CDC28* gene, then transformed into both the YPH500 (*UBR1 ura3*) and JD15 (*ubr1Δ ura3*) strains of yeast, and clones in which the plasmid had integrated into the endogenous *CDC28* gene were identified for further analysis.

**Drosophila.** CactDN is a mutant version of the Cactus protein in which serines 74, 78, and 116 have been converted to alanine residues [80], rendering the protein insensitive to Toll receptor dependent phosphorylation, and subsequent ubiquitination and degradation. This protein binds constitutively to the Dorsal protein, resulting in its sequestration in the cytoplasm. Females expressing CactDN in their germlines produce embryos with a dominant-negative dorsalized phenotype. We reasoned that the expression of a PND-tagged version of CactDN in otherwise wild-type females would result in larvae that were dorsalized when early embryogenesis progressed in darkness, and normalized when embryos developed under illumination, owing to the degradation of CactDN, which would enable Dorsal to come under the regulation of endogenous wild-type Cactus protein. We generated a PND-tagged version of Cactus as follows:

First, the two oligonucleotides: /5'Phos/TGGCCGCTTGGCTACTACACTTGAACG and /5'Phos/CCTCTTAGCCTTAGCACAAGATGTAAGG were used for high fidelity inverse PCR-mediated *in vitro* mutagenesis of the plasmid *pPW-UBI-R-phLOV2-HA-URA3*, in order to eliminate the NotI site at the junction between the ubiquitin open reading frame and the arginine codon preceding the LOV2 domain. This yielded plasmid *pPW-UBI-NotI<sup>minus</sup>-R-phLOV2-HA-URA3*.

Next, the two oligonucleotides: 5'GATCGAGCGGCCGCAAAATGCAGATTTTCGTCAA GACTTTGACCGG and 5'GATCGAGGATCCCCTCCTAAAAATGCAGCGTAATCTGGA ACATCG were used for high fidelity PCR, using *pPW-UBI-R-phLOV2-HA-URA3* as a template, in order to generate an amplification product comprising the ubiquitin open reading frame, arginine codon, LOV domain and HA tag. This amplification product was restriction-digested with NotI and BamHI and the DNA fragment encoding the PND was purified.

The two oligonucleotides: 5'ACGTACGGATCCGAGCCCAACAAAAGCAGCGGAGGC and 5'ACGTACGCTAGCTCAGGCAACTGTCATGGGATTGCCACCG were used for high fidelity PCR, using a plasmid carrying the open reading frame corresponding to CactDN [80] as a template. The amplification product was then restriction digested with BamHI and NheI and the DNA fragment encoding CactDN purified.

Finally, the *Drosophila* germline expression vector *pUASp* [110], was digested with NotI and XbaI and the larger fragment purified. This fragment was combined with the *NotI/BamHI PND* fragment and the *BamHI/NheI CactDN* fragment, generating *pUASp-PND-HA-CactDN*, in which the *UBI* gene, and the R-phLOV2, HA tag, and CactDN coding sequences were present in-frame and under the transcriptional control of the *Gal4* upstream activating sequences enhancer element. This plasmid was introduced into the *Drosophila* genome by conventional P-element-mediated transposition following microinject of embryos at Rainbow Transgenics, Inc.

The psd (for photosensitive degron) [56] is comprised of the LOV2 domain from *Arabidopsis thaliana* phototropin 1 protein combined with a 23 amino acid in length unstructured peptide from a synthetic degron [83] that was derived from a natural degron present in murine ornithine decarboxylase [84]. In order to generate a version of CactDN whose degradation was under the control of the psd, the two oligonucleotides: 5'ACTGACGGATCCGAGAGGTG AACAAAAGTTGATTTCTGAAGAAGATTTGAACGGTG and 5'CATGACACTAGTTA TTGGAAGTACAAGTTTTCAGAACCAGCC. were used for high fidelity PCR, using a plasmid carrying sequences encoding a copy of the myc epitope tag in frame with the psd [56] (a kind gift of Christof Taxis). The amplification product was restriction digested with BamHI and SpeI and ligated to plasmid pUASp that had been restriction digested with BamHI and XbaI, resulting in plasmid *pUASp-myc-psd*. Next, the two oligonucleotides: 5'ACGTGATCG CGGCCGCAAAATGCCGAGCCCAACAAAAGCAGCGGAGGC and 5'GATCGAGGATC CGCAACTGTCATGGGATTGCCACCGTTG were used for high fidelity PCR, using a plasmid carrying the open reading frame corresponding to CactDN as a template. The amplification product was then restriction digested with NotI and BamHI and ligated to similarly digested *pUASp-myc-psd*, yielding plasmid *pUASp-CactDN-psd*, in which the CactDN coding sequences, the myc epitope and the psd were present in-frame and under the transcriptional control of the *Gal4* upstream activating sequences enhancer element. This plasmid was introduced into the *Drosophila* genome by conventional P-element-mediated transposition following microinject of embryos at Rainbow Transgenics, Inc.

The four amino acid sequence arg-arg-arg-gly (RRRG), fused to the C-terminus of a protein results in rapid proteasomal degradation of the protein in mammalian cells [26]. The combination of a mutated variant of the *Avena sativa* LOV2 core domain with the RRRG peptide resulted in the formation of a light-dependent degron, referred to as the B-LID domain [57].

In order to generate a version of CactDN whose degradation was under the control of the B-LID domain, the two oligonucleotides: 5'GACGAGCTGGATCCGACGCGTTTCTTGGC TACTACACTTGAACG and 5' GCGGATCGTCTAGACTAACCTCGCCGCCTTGCCGCCT CATC. were used for high fidelity PCR, using a plasmid carrying sequences encoding the B-LID domain, *pBMN HAYFP-LOV24* (Addgene #49570) [57]. The amplification product was purified and restriction digested with BamHI and XbaI and ligated to similarly digested *pUASp*, yielding plasmid *pUASp-B-LID*. Subsequently, the NotI/BamHI digested DNA fragment encoding CactDN that is described above was ligated to similarly digested *pUASp-B-LID*, yielding *pUASp-CactDN-B-LID*, in which the *CactDN* and the *B-LID* domain sequences have been fused in-frame and have been placed under the transcriptional of the *Gal4* upstream activator sequences enhancer element. This plasmid was introduced into the *Drosophila* genome by conventional P-element-mediated transposition following microinject of embryos at Rainbow Transgenics, Inc.

## Examination of yeast phenotypes resulting from light exposure

For yeast grown on solid medium, individual colonies carrying the constructs being tested were suspended in 100 μl of sterile water and serial 10-fold dilutions were generated. 5 μl droplets of each of the serial dilutions were applied to a petri dish containing appropriate selective media. The dilution that had resulted in the deposition of a single layer of separated cells, as observed under a dissecting microscope at 80X, was used to determine the phenotypic consequences of light exposure. Over the course of these studies, it became apparent that in order to achieve reliable determinations of the light-dependent phenotypes of yeast carrying the PND-tagged Ura3p and Cdc28p constructs on plates, it was necessary to plate those yeast strains in a single layer on the agar surface. These observations indicated that when cells were plated at higher density, in more than a single layer, the cells at the surface shielded underlying cells from light exposure. Plates were incubated at 30˚C (cover side up) at a distance of 6 cm from a Blue 225 LED 13.8 Watt/110 Volt Square Grow Light Panel (LEDwholesalers 2501BU) and examined after 24 and 48 hours of incubation. Control replica plates were incubated in a light/tight container in the same incubator. To test the effects of red light on yeast cell phenotypes, similar plates were grown under a Red 225 LED 13.8 Watt/110 Volt Square Grow Light Panel (LEDwholesalers 2501RD) and examined after 24 and 48 hours, again in tandem with duplicate plates that were grown in a light/tight container.

## Western blot analysis of PND-Cdc28p expressed in yeast

For yeast grown in liquid culture, overnight cultures of cells were grown up in 6 mls of SD medium with appropriate selection. The next morning, a small volume of the culture was diluted into 200 mls of selective SD medium to achieve an OD between .06 and .08. The 200 mls were divided into four 50-ml cultures in 250 ml baffled Erlenmeyer flasks and grown with shaking (200 rpm) at 30˚ C in the dark for 4–6 hours. The cultures were combined and the OD measured. The experiment was started when the OD was between 0.1 and 0.2. At time 0 an initial sample was taken that corresponded to ~5 ODs, and the remaining culture was divided evenly between the four flasks. For the time course experiments, the two flasks containing the cultures to be grown in the dark were covered in foil. To provide illumination for the cultures to be grown in the light, a White and Blue High-Power LED Aquarium light, (2518W+B, LEDwholesalers) was suspended 18 cm above the platform on which the culture flasks were shaken. Only the blue LED bulbs were used during culture growth. At each time point, the cultures in the two flasks receiving the same treatment were combined and the OD determined. A volume corresponding to 2.5 ODs was removed and the remaining culture evenly distributed between the two flasks and returned to shaking. For the cycloheximide experiments, after time 0 all the flasks were either kept in the dark or they were all illuminated, with two of the flasks receiving cycloheximide to a concentration of 100 mM. For these experiments, the OD was determined only at time 0 and at the end point (60 minutes). Collected samples were centrifuged at 1700 RPM (4˚ C) in a 50 ml conical tube for 3 minutes. The supernatant was discarded, and the cell pellets were resuspended in 1 ml of water and centrifuged for 3 minutes at 1600 rpm in an Eppendorf tube. The supernatant was removed and the pelleted cells were frozen in liquid nitrogen.

Protein was extracted from the samples following the alkaline lysis method of Kushnirov [111]. For each 2.5 ODs of material, the pellet was solubilized in 100 μl $H_2O$. 100 μl of 0.2M NaOH was added and the contents gently mixed. The tubes were incubated for 5 minutes at room temperature, then centrifuged for 10 seconds at 13,200 rpm in an Eppendorf centrifuge. The supernatant was discarded and the pellet was resuspended in 75 μl of 1X Laemmli gel sample buffer. The sample was then boiled for 3 minutes and spun at 13,200 rpm for 5 minutes.

The supernatant was removed to a new tube and frozen in liquid nitrogen. Protein concentration were determined using the Bio-Rad Protein Assay reagent (Cat# 500–0006, Bio-Rad Laboratories, Hercules, CA). For Western blots, 75 or 100 µg of protein were loaded per lane.

After blotting, the top and bottom half of the gels were separated just below the 49 kD molecular weight marker. The top half was incubated in rabbit anti-HA epitope tag (Rockland antibodies & assays, Cat. #600-401-384S) at a dilution of 1:5,000, while the bottom half was incubated in mouse anti-Glyceraldehyde-3-phosphate dehydrogenase (GAPDH/GA1R) from ThermoFisher Scientific (Cat. # MA5-15738) at a dilution of 1:10,000. Respective secondary antibodies were HRP-conjugated goat anti-rabbit IgG and goat anti-mouse IgG polyclonal antibodies from Jackson Laboratories (Cat. #s 111-035-003 and 115-035-003) both used at a dilution of 1:10,000. All antibody incubations were carried out overnight at 4˚C.

### Examination of *Drosophila* cuticular phenotypes and hatch rates resulting from light exposure

For studies of the effects of illumination on cuticular phenotypes and on hatch rates associated with photo-N-degron-mediated protein degradation in embryos, females expressing the degron-tagged CactDN pUASp [110] constructs under the control of the *nanos-Gal4:VP16* [81] transcriptional driver element were collected and introduced with males into egg collection cages on yeasted apple juice agar plates. For light-exposed embryos, females were allowed to lay eggs for 1 hour in the dark at 25˚C, at which time the plates were removed and transferred cover side up to a shelf in another 25˚C incubator at a distance of 6 cm from a Blue 225 LED 13.8 Watt/110 Volt Square Grow Light Panel (LEDwholesalers 2501BU). Embryos were allowed to develop for at least 48 hours and cuticles prepared from unhatched eggs present on the plates. Females of the same genotype were also allowed to lay eggs in the dark and embryos allowed to develop for at least 48 hours, in order to assess the phenotypes of embryos in which the degron-tagged CactDN proteins had not been exposed to light, also by preparing and examining cuticle preparations from unhatched eggs.

Larval cuticles were prepared according to Van der Meer [112]. Dorsal/Ventral phenotypes of embryos were classified as described in Roth et al. [69], with modifications as follows. DO embryos are completely dorsalized, exhibiting only dorsally-derived cuticle with fine hairs all around their DV circumference. D1 embryos are strongly dorsalized, exhibiting only dorsal and dorsolateral (Filzkörper material) structures. D2 embryos are moderately dorsalized, exhibiting dorsal, dorsolateral, and ventrolateral structures. These embryos displayed Filzkörper or Filzkörper material as well as narrower than normal bands of ventral denticles. D3 embryos are weakly dorsalized and display Filzkörper and ventral denticles of normal width. These embryos exhibited a twisted, or tail-up/U-shaped phenotype, consistent with a disruption of mesoderm tissue, the ventral-most pattern element in the embryo and often exhibited disruptions of the head skeleton. UH (for unhatched) refers to embryos exhibiting apparently normal cuticular pattern elements, but which failed to hatch from the egg.

### Live imaging of embryos

Embryos were collected one and a half hours after egg deposition, dechorionated by hand, and staged in halocarbon 27 oil. Embryos expressed Dorsal-GFP under the control of the endogenous *dorsal* gene transcriptional regulatory elements [85] together with the degron-tagged CactDN construct expressed under the control of the *mat-α4-tub-Gal4:VP16* transcriptional driver element [108]. All the embryos were prepared under red filtered light (red film—Neewer, 10087407) to avoid possible degradation of degron-tagged CactDN by light emitted by microscopes. Embryos at nuclear cycle (nc) 11 were mounted in Halocarbon 27 oil (Sigma-

Aldrich) between a glass slide and a coverslip using glue dissolved in heptane with folded double-sided tape placed between the slide and the coverslip. Embryos were imaged on a Zeiss LSM 800 confocal microscope using a 25x oil immersion objective. To increase time-resolution, we used a 0.8 digital magnification. Images were captured at 512 x 512 pixel resolution with the pinhole set to a diameter of 50 mm. At each time point, a stack of 30 z-plane images separated by 0.3 mm were captured, spanning the nuclear layer.

To test efficiency of loss of activity of the degron-tagged CactDN constructs upon blue laser illumination, 488nm light ("blue laser") at either low (2–3.1%) or high (8.6–10%) laser power was applied to the experimental embryos able to stimulate GFP fluorescence emission (i.e. 3.1%) as well as to induce both GFP fluorescence as well as degron-tagged Cactus degradation (i.e. 10%). While for control embryos, 488 nm laser light also was applied but only at low power; a setting that is able to stimulate GFP fluorescence emission but is not expected to contribute substantially to the loss of degron-tagged Cactus (or directs very slow degradation relative to high power illumination). Unless otherwise noted, embryos exposed to blue laser (488 nm) illumination were imaged under three conditions as outlined in Fig 9A: (i) imaging was initiated at nc12 using the 488nm laser at 3.1% power (or 2% for CactDN-B-LID); (ii) 10min later (nc13) power was increased to 10% (or 8.6% for CactDN-B-LID) and applied for a period of 20min; and after a resting period of 35min, imaging was reinitiated at low power (2–3.1%) until late nc14/gastrulation. S1, S2, S4, S6 Movies are a compilation of these three imaging sessions. For the low-blue light condition (S3, S5, and S7 Movies), the embryos were imaged with low power 488nm laser light sufficient to illuminate the Dorsal-GFP but expected to have little impact on photosensitive-degrons, from the onset of nc12 to nc14 and after 35min rest, approximately at late nc14/gastrulation an image was captured again. S8 Movie, a compilation of two imaging sessions, was initiated (i) earlier at nc12, starting with blue laser (488 nm) illumination for 20 min at 10% laser power and then followed by imaging (ii) at 488 nm at 3.1% laser power until nc14. Emission signal for low laser power was collected from 495–541 nm and for high laser illumination from 400-541nm. Movies displaying Z-stack projections (scanned area: S1–S5 movies 0.8x and S6-S8 1.2x) of the Dorsal-GFP gradient were obtained using Fiji/ImageJ software. Images in Figs 9 and 10 were generated as selected stacks of 15–20 z-plane images (planes with high background were avoided) from timed frames of each equivalent movie. The laser power can fluctuate with any set-up over time and thus must be empirically defined for each set of experiments (e.g. The first experiments that were carried out with CactDN-B-LID/Dorsal-GFP required less laser power for both visualization of Dorsal-GFP and elimination of CactDN-B-LID).

## Western blot analysis of Cactus constructs expressed in *Drosophila* embryos

For Western blot analysis of Cactus proteins, eggs laid by transgenic females expressing the *CactDN-B-LID* or *PND-HA-CactDN* construct under the transcriptional control of the *nanos-Gal4:VP16* [81] transcriptional driver element were collected at 2–4 hours after egg deposition on yeasted apple juice/agar plates. Eggs were laid and incubated in either dark condition or blue light condition at 25˚C until collection, in ambient light. Following collection, eggs were dechorionated in 50% Chlorox bleach, transferred to 1.7 ml microcentrifuge tubes, and homogenized with a microcentrifuge tube-compatible pestle in roughly equal volume of lysis buffer (25 mM Tris, pH 7.5 /0.15 M NaCl /0.3% NP-40 /1mM EDTA/ 1mM EGTA /0.2mM N-ethylmaleimide, containing protease inhibitors [Pierce Protease Inhibitor Tablets, EDTA-free, Pierce Biotechnology, Rockford, IL]). Protein concentrations in the homogenates were determined using the Bio-Rad Protein Assay reagent. For each embryo extract, a volume corresponding to total protein of 200 µg for the CactDN-B-LID sample and 50 µg for the

PND-CactDN was subjected to SDS polyacrylamide gel electrophoresis. Following electroblotting to nitrocellulose membranes, the CactDN-B-LID blot was incubated with monoclonal primary antibodies against either Cactus (1/500) (Mouse Monoclonal 3H12, DSHB, Hybridoma deposited by Steward, R.) or Tubulin (1:1,000) (Mouse Monoclonal clone DM1A, Product# T6199, Sigma-Aldrich, St. Louis, MO). The PND-HA-CactDN blot was incubated with anti-HA (1:1000)(Mouse Monoclonal 16B12, Prod# MMS-101P, Covance Inc., Emeryville, CA). Blots were washed and incubated with Peroxidase-conjugated Goat Anti-Mouse IgG (1:10,000 for CactDN-B-LID and 1:5,000 for PND-HA-CactDN) (Code# 115-035-003, Jackson ImmunoResearch Laboratories, West Grove, PA). Signal was detected using the SuperSignal West Pico Chemiluminescent Substrate (Prod# 34080, Pierce Biotechnology, Rockford, IL). The PND-HA-CactDN blot was imaged using a C-DiGit blot scanner and Image Studios Software (LI-COR Biosciences).

## Supporting information

**S1 Movie. Dorsal-GFP entry into nuclei in the absence of degron-tagged CactDN under the conditions of laser illumination used in this study.** A single embryo expressing Dorsal-GFP was imaged from nuclear cycle (nc) 12 until gastrulation (st.6), demonstrating the formation of a normal ventral-to-dorsal gradient of nuclear accumulation in the absence of CactDN expression. In brief, the embryo was imaged under three conditions as outlined in Fig 9A: (i) imaging was initiated at nc12 using the low power (3.1%) 488nm laser; (ii)10min later (nc13) high power (10%) 488nm was applied for a period of 20min; finally, after a resting period of 35min, imaging was reinitiated at low power (3.1%) 488nm, extending until gastrulation. The movie is a sequential compilation of these 3 imaging sessions taken over time, and due to lack of CactDN serves as a control for the movies of embryos expressing the various degron-tagged versions of CactDN. Snapshots from the movie are shown in Fig 9B–9B‴.
(ZIP)

**S2 Movie. Blue laser light induces nuclear accumulation of Dorsal-GFP expressed together with PND-HA-CactDN.** The movie shows an embryo expressing Dorsal-GFP and PND-HA-CactDN imaged using the same conditions as described for S1 Movie from nc12 to nc14/gastrulation; importantly, including 20 min high power blue light (i.e. 488nm, 10%) illumination used to initiate degron-mediated loss of PND-HA-CactusDN activity at nc13/14, resulting in Dorsal-GFP nuclear localization. Snapshots from the movie are shown in Fig 9C–9C‴.
(ZIP)

**S3 Movie. Perturbation of nuclear localization of Dorsal-GFP expressed together with PND-HA-CactDN under low intensity light.** The movie shows an embryo expressing Dorsal-GFP together with PND-HA-CactDN fusion protein, imaged under low power blue light only (488 nm, 3.1%) initiating at nc12 and continuing until late nc14/gastrulation over a period of ~75 min. Snapshots from the movie are shown in Fig 9D–9D‴.
(ZIP)

**S4 Movie. Blue laser light induces nuclear accumulation of Dorsal-GFP expressed together with CactDN-B-LID.** The movie shows an embryo expressing Dorsal-GFP and CactDN-B-LID using the same conditions described for S1 Movie, from nc12 to nc14/gastrulation, importantly including 20 min high power blue light illumination (488nm, 8.6%) to initiate degron-mediated loss of CactDN-B-LID activity at nc13/14, resulting in Dorsal-GFP nuclear localization. Snapshots from the movie are shown in Fig 9E–9E‴.
(ZIP)

**S5 Movie. Perturbation of nuclear localization of Dorsal-GFP expressed together with CactDN-B-LID under low intensity light.** The movie shows an embryo expressing Dorsal-GFP together with CactDN-B-LID, imaged under low power blue light only (488 nm, 2%) initiating at nc12 and continuing until nc14/gastrulation. Snapshots from the movie are shown in Fig 9F–9F‴.
(ZIP)

**S6 Movie. Laser illumination of live embryos expressing CactDN-psd induces transient cyclical nuclear accumulation of Dorsal-GFP (low mag).** The movie shows an embryo expressing Dorsal-GFP together with CactDN-psd from nc12 up to nc14. In brief, embryos were imaged under two conditions: (i) first, imaging was initiated at nc12 using the low power 488nm laser; (ii) after mitotic division (nc13), high power 488nm laser light was applied for a period of 20min. Importantly, high power blue light (488nm) illumination initiated degron-mediated loss of CactDN-psd activity during nc13, resulting in transient Dorsal-GFP nuclear localization just before the onset of nuclear mitosis. The movie is a compilation of these 2 imaging sessions taken over time. Snapshots from the movie are shown in Fig 10A–10A‴.
(ZIP)

**S7 Movie. Dorsal-GFP expressed together with CactDN-psd fails to accumulate in nuclei under low intensity light.** The movie shows an embryo expressing Dorsal-GFP together with CactDN-psd imaged under low power 488 nm laser light, 3.1% from nc12 to nc14. Snapshots from the movie are shown in Fig 10B–10B‴.
(ZIP)

**S8 Movie. Laser illumination of live embryos expressing CactDN-psd induces transient cyclical nuclear accumulation of Dorsal-GFP (higher mag).** The movie shows an embryo expressing Dorsal-GFP together with CactDN-psd which was exposed to high power blue light (488nm, 10%) for 20 min, initiating at nc12 and extending into nc13, and subsequently imaged with low power light (488 nm, 3.1%) until the end of nc13. Snapshots from the movie are shown in Fig 10C–10C‴.
(ZIP)

## Acknowledgments

We thank Winslow Briggs, Arko Dasgupta, Neta Dean, R. Jürgen Dohmen, Jay C. Dunlap, Arlen Johnson, Karim Labib, Jennifer J. Loros, Christof Taxis, Tong-Seung Tseng, Thomas J. Wandless, and Alexander Varshavsky for reagents and/or technical advice. We thank Katie Sieverman, Emily Flynn, and Leslie Dunipace for technical assistance.

## Author Contributions

**Conceptualization:** David S. Stein.

**Data curation:** Leslie M. Stevens, Goheun Kim, Theodora Koromila, David S. Stein.

**Formal analysis:** Goheun Kim, Theodora Koromila.

**Funding acquisition:** Angelike Stathopoulos, David S. Stein.

**Investigation:** Leslie M. Stevens, Goheun Kim, Theodora Koromila, John W. Steele, James McGehee, David S. Stein.

**Methodology:** Leslie M. Stevens, Goheun Kim, Theodora Koromila, John W. Steele, James McGehee, David S. Stein.

**Project administration:** Angelike Stathopoulos, David S. Stein.

**Resources:** Goheun Kim, John W. Steele, James McGehee, Angelike Stathopoulos, David S. Stein.

**Supervision:** Angelike Stathopoulos, David S. Stein.

**Validation:** Leslie M. Stevens, Goheun Kim, Theodora Koromila, James McGehee, Angelike Stathopoulos, David S. Stein.

**Visualization:** Leslie M. Stevens, Goheun Kim, Theodora Koromila, David S. Stein.

**Writing – original draft:** David S. Stein.

**Writing – review & editing:** Leslie M. Stevens, Goheun Kim, Theodora Koromila, James McGehee, Angelike Stathopoulos, David S. Stein.

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
