## [Decision Letter · Decision Letter 0]

27 Jan 2021

Dear Dr Stein,

First, let me profusely apologize for the time this has taken. I acted on it as soon as it was assigned to me.  But getting good reviewers proved difficult indeed.That said, we  now have very positive reviews from two excellent reviewer. Both urge ACCEPT PENDING MINOR REVISION.  This is good news, happy news! Congrats!  That said, I really urge (insists/) you to take their comments quite seriously and revise the paper along the lines they suggest!  I truly look forward to seeing the revision.  Now for the boiler plate:

Thank you very much for submitting your Research Article entitled 'Light-dependent N-end rule-mediated disruption of protein function in Saccharomyces cerevisiae and Drosophila melanogaster' to PLOS Genetics.

The manuscript was fully evaluated at the editorial level and by independent peer reviewers. The reviewers appreciated the attention to an important topic but identified some concerns that we ask you address in a revised manuscript.

We therefore ask you to modify the manuscript according to the review recommendations. Your revisions should address the specific points made by each reviewer.

[LINK]

Yours sincerely,

R. Scott Hawley

Associate Editor

PLOS Genetics

Chengqi YI

Section Editor: Methods

PLOS Genetics

Reviewer's Responses to Questions

**Comments to the Authors:**

Reviewer #1: The paper by Stevens et al. reports the development of a new optogenetic tool to study protein function in vivo: the photo-N-degron (PND). This N-terminal tag is based on an N-terminal ubiquitin moiety, followed by a blue-light sensitive LOV domain (phLOV2 domain from Avena sativa) placed immediately C-terminal to an amino terminal arginine residue. In the presence of light the LOV domain undergoes a conformational change that exposes the Arg to the N-end rule degradation pathway.

The authors test the effectiveness of PND in S. cerevisae and D. melanogaster, as well as two other previously reported degrons, B-LID and psd, in D. melanogaster embryos. They show that PND displays light-dependent degradation in vivo, with effects observed as short as 1 min after illumination. Using the yeast system they fuse the PND tag to three different proteins, namely Ura3p, EmRFP and the cell cycle Cdc28 protein. They show that for all three, protein degradation is dependent on illumination and on the presence of ubiquitin. In Drosophila, they use a dominant-negative form of the Cactus inhibitor to test the effects of PND, B-LID and psd degrons.

These are very carefully controlled experiments, which generate and test new tools for fine temporal control of gene expression. In addition, they set the basis for the future development of optogenetic tools to study spatially restricted protein function, with great impact in genetic studies in general. Therefore, I look forward to see this manuscript published. I have only a few minor suggestions, as listed below.

line 199. The authors cite for the first time the use of the yeast UBR1 ura3 mutant strain. It would be helpful to add a sentence explaining the overall characteristics of that strain, which is used several times throughout the manuscript. This would avoid having to report back to the original paper for basic information. Alternatively, the same type of information could be added to the methods section.

line 315. The authors examine whether PND-dependent degradation occurs post-translationally or by degradation of nascent peptides, the latter process limiting the use of the tool for the study of loss-of-function phenotypes. They do so by growing PND-HA-Cdc28 expressing cells in the presence of cycloheximide and find that protein degradation in the presence of light is still observed, suggesting a post-translational mechanism. While this seems true for cdc28, different effects could, in principle, hold for other genes and must be acknowledged. That is, is it possible that the test performed in the presence of cycloheximide only rules out a co-translational effect for Cdc28p but not necessarily for other genes? To discuss this possibility prepares the reader for limitations they might encounter should they consider using this tool for their own protein of interest.

For their Drosophila studies, Stevens et al use a dominant negative form of the IkB inhibitor Cactus that is insensitive to Toll-induced degradation (cactDN), generating a dorsalizing phenotype. If degradation is effective, the embryonic phenotype is reverted by illumination, a smart way to study the utility of the degrons to access protein function in vivo. They generate several insertion lines expressing CactDN-degrons and cite that they have some line-to-line variability due to different expression levels (line 407). This generates diversity in the number and severity of dorsalized embryos grown in the dark. Since their table containing results on the viability of these embryos gathers the different lines, I am curious about how this variability in expression levels may alter the effectiveness of the degron for the study of protein function. It would be interesting if they could add this information.

The authors cite (line 541):

"our studies of the ability of the cactDN degron to control the nuclear accumulation of Dorsal-GFP indicate that ...PND and B-LID are capable of mediating temporally-specific elimination of Cact within the blastoderm embryo"

Temporal control is certainly a strong characteristic shown for the degrons above and should be very useful for developmental studies. Since they have used the same tag for several yeast proteins, I wonder whether they could compare the level of temporal control attained with those proteins as well. If they could explore a bit more this issue it would be useful in giving a sense of the generality of this phenomenon.

Discussion, line 602: The authors consider differences in intrinsic protein stability an important property to take into account when devising a degron to study loss-of-function phenotypes for a protein of interest. In the same line of reasoning, could they discuss whether and how expression levels would impact degron effectiveness?

This information may be useful especially considering the advent of CRISPR technology. Optogenetic tags that were tested with expression constructs by Stevens et al may be useful to tag endogenous genes, as considered in the Discussion. However, one limitation that can be foreseen is the expression levels obtained in tagging endogenous genes as compared to overexpression constructs. Do the authors have information on the expression levels of their tagged constructs versus endogenous expression? The western blot in Fig 8 shows slightly higher levels of Cact-BLID as compared to endogenous Cact levels. On the other hand, in their paper using a Dl-degron they generate much higher levels of construct expression compared to endogenous. Do they think this is an important parameter to be considered for different proteins?

Methods and legends:

About the GAL4 drivers used in the Drosophila experiments: in line 360 the authors cite the use of the nanos-GAL4 (Table 1). Is this used for Fig 7A also?

In Fig 7B and Table 2 it is also unclear which GAL4 is used. Since nothing is said in legends, one assumes it is the same GAL4 driver as the previous figures. However, in Methods they state that for embryo illumination they use either nanos or mat GAL4 drivers (line 929) for viability and cuticle phenotypes, but do not specify in each experiment which GAL4 driver was used. Please specify. For western blot experiments I could not find information as to which GAL4 driver was used.

Reviewer #2: In this manuscript, the authors describe the development of photo-N-degron (PND), an N-terminal blue light-inducible degron, for the induction of loss-of-function phenotypes in yeast and in Drosophila. They tested the feasibility of PND both by phenotypic and biochemical (westernblots) analysis. In the first part of the manuscript, the authors describe the use of PND in three examples in yeast. Here, they also show that the degradation of PND is dependent on the N-end rule degradation pathway. In the second part, the authors tested the functionality of PND in a multi-cellular organism, Drosophila. Specifically, they make a fusion of PND with Cactus[DN] which cases a dominantly dorsalizing phenotype in Drosophila embryos. Both in the yeast and Drosophila assays, the authors demonstrate that PND can be used for blue light-induced protein degradation. In the Drosophiila part, the authors also compared the PND to two other degrons, PSD and B-LID, which are carboxy-terminal light-induced degrons. While the Cact-PSD fusion does not appear to respond to blue-light illumination, the Cact-B-Lid fusion responds very well, maybe even better than PND, to blue light. The authors also examine in great detail the effect of the Cact[DN] fusion proteins on the nuclear localization of Dorsal in light and dark conditions.

Overall, this is an interesting study which introduces an N-terminal blue light-inducible degron for optogenetic studies of protein function in vivo. It increases the toolbox for these kind of studies and increases the flexibility of researchers to either target proteins at the N- or C-terminus. The experimental procedures used in this study are well explained and the conclusions are supported by the data. I have only a few minor concerns/questions to further improve the manuscript.

1. My major concern is the length of this manuscript. It is way too long and should be shortened. There are many redundant parts where the authors repeat themselves over and over. Figure 7 and Tables 1 and 2 contain the same content. Either the tables or the figure could be removed. The Discussion is very good, but also too long.

2. In Figure 2, it wasn't clear to me why a ura3 mutant strain was used, at least this was mentioned in the text. In the actual figure and legend, ura3 was not mentioned. Please clarify.

3. In Figure 7, the authors use D4 as an additional class of dorsalized embryos, but don't explain it in the text. In line 395, the introduce UH as abbreviation for unhatched embryos, but don't use it the figure. Is it possible that D4 corresponds to UH?

**Have all data underlying the figures and results presented in the manuscript been provided?**

Reviewer #1: Yes

Reviewer #2: Yes

PLOS authors have the option to publish the peer review history of their article (what does this mean?). If published, this will include your full peer review and any attached files.

Reviewer #1: **Yes: **Helena Araujo

Reviewer #2: **Yes: **Andreas Bergmann

---

## [Editor Report · Decision Letter 1]

12 Apr 2021

Dear Dr Stein,

We are pleased to inform you that your manuscript entitled "Light-dependent N-end rule-mediated disruption of protein function in Saccharomyces cerevisiae and Drosophila melanogaster" has been editorially accepted for publication in PLOS Genetics. Congratulations! (Thank you for providing us with a predictably thorough and scholarly response and revision.)

Yours sincerely,

R. Scott Hawley

Associate Editor

PLOS Genetics

Chengqi YI

Section Editor: Methods

PLOS Genetics

Comments from the reviewers (if applicable):

**Data Deposition**

http://datadryad.org/submit?journalID=pgenetics&manu=PGENETICS-D-20-01548R1

**Press Queries**

---

## [Editor Report · Acceptance letter]

12 May 2021

PGENETICS-D-20-01548R1 

Light-dependent N-end rule-mediated disruption of protein function in Saccharomyces cerevisiae and Drosophila melanogaster 

Dear Dr Stein, 

We are pleased to inform you that your manuscript entitled "Light-dependent N-end rule-mediated disruption of protein function in Saccharomyces cerevisiae and Drosophila melanogaster" has been formally accepted for publication in PLOS Genetics! Your manuscript is now with our production department and you will be notified of the publication date in due course.

With kind regards,

Katalin Szabo

PLOS Genetics

On behalf of:
